# Mechanosensitive pore opening of a prokaryotic voltage-gated sodium channel

**Peter R Strege[1], Luke M Cowan[1], Constanza Alcaino[1], Amelia Mazzone[1], Christopher A Ahern[2], Lorin S Milescu[3]\*, Gianrico Farrugia[1,4]\*, Arthur Beyder[1,4]\***

[1]Enteric Neuroscience Program (ENSP), Division of Gastroenterology & Hepatology, Department of Medicine, Mayo Clinic, Rochester, United States; [2]Department of Molecular Physiology and Biophysics, University of Iowa, Iowa City, United States; [3]Department of Biology, University of Maryland, College Park, College Park, United States; [4]Department of Physiology and Biomedical Engineering, Mayo Clinic, Rochester, United States

**\*For correspondence:**
lorinsmilescu@gmail.com (LSM);
farrugia.gianrico@mayo.edu (GF);
Beyder.Arthur@mayo.edu (AB)

**Competing interest:** The authors declare that no competing interests exist.

**Abstract** Voltage-gated ion channels (VGICs) orchestrate electrical activities that drive mechanical functions in contractile tissues such as the heart and gut. In turn, contractions change membrane tension and impact ion channels. VGICs are mechanosensitive, but the mechanisms of mechanosensitivity remain poorly understood. Here, we leverage the relative simplicity of NaChBac, a prokaryotic voltage-gated sodium channel from *Bacillus halodurans*, to investigate mechanosensitivity. In whole-cell experiments on heterologously transfected HEK293 cells, shear stress reversibly altered the kinetic properties of NaChBac and increased its maximum current, comparably to the mechanosensitive eukaryotic sodium channel $Na_V1.5$. In single-channel experiments, patch suction reversibly increased the open probability of a NaChBac mutant with inactivation removed. A simple kinetic mechanism featuring a mechanosensitive pore opening transition explained the overall response to force, whereas an alternative model with mechanosensitive voltage sensor activation diverged from the data. Structural analysis of NaChBac identified a large displacement of the hinged intracellular gate, and mutagenesis near the hinge diminished NaChBac mechanosensitivity, further supporting the proposed mechanism. Our results suggest that NaChBac is overall mechanosensitive due to the mechanosensitivity of a voltage-insensitive gating step associated with the pore opening. This mechanism may apply to eukaryotic VGICs, including $Na_V1.5$.

## Editor's evaluation

This important study presents a technically impressive and carefully controlled biophysical study of the nature of the mechanosensitivity of voltage-gated sodium channels. The identification of a mechanosensitive step with little voltage sensitivity is convincing, and the proposal of a swinging door mechanism for the intracellular gate is plausible. It is expected to be of interest to scientists studying sodium channels and the physical basis of mechanosensitivity in electrophysiology.

## Introduction

Electrically excitable tissues with mechanical functions like the heart and gut using VGICs to generate electrical activity, which drives mechanical activity via electro-mechanical coupling (*Hille, 2001*). Conversely, mechanical movements change membrane tension and impact electrical function in a process called mechano-electrical feedback (*Kohl et al., 2005*), which relies on specialized

mechanically-gated ion channels, such as TREK (*Brohawn et al., 2014*) and Piezo (*Ranade et al., 2015*). However, studies dating back nearly 40 years suggest that VGICs are also mechanosensitive and thus may directly contribute to mechano-electrical feedback (*Conti et al., 1982*; *Conti et al., 1984*; *Hao et al., 2013*; *Strege et al., 2003*; *Terakawa, 1983*). Indeed, most VGIC families display mechanosensitivity, including sodium (Na$_V$) (*Morris and Juranka, 2007*), potassium (K$_V$) (*Gu et al., 2001*; *Schmidt et al., 2012*), calcium (Ca$_V$) (*Farrugia et al., 1999*), proton (H$_V$) (*Pathak et al., 2016*), and cyclic nucleotide-gated (HCN) (*Lin et al., 2007*) channels. An important mechanistic advance was made in a recent study that showed that Kv channels are exquisitely mechanosensitive in their opening transition (*Schmidt et al., 2012*).

Mechano-electrical feedback via VGICs can play a distinct physiological role. Unlike the specialized mechano-gated channels whose activation is generally voltage-insensitive, mechanosensitive VGICs create a 'voltage-informed' mechano-electrical feedback (*Gaub et al., 2020*; *Hao et al., 2013*). Perhaps the best example is the voltage-gated sodium channel Na$_V$1.5, responsible for the upstroke of cardiac action potentials (*Gellens et al., 1992*). Given the heart's role as a pump, Na$_V$1.5 is a natural target for mechanosensitivity investigations, and several studies showed that macroscopic Na$_V$1.5 currents are mechanosensitive (*Beyder et al., 2010*; *Morris and Juranka, 2007*). Interestingly, disease-associated Na$_V$1.5 mutations (channelopathies) can affect mechanosensitivity (*Banderali et al., 2010*; *Beyder et al., 2014*; *Strege et al., 2018*). Furthermore, lipid-permeable anesthetics and amphipathic drugs such as ranolazine that target Na$_V$1.5 inhibit its mechanosensitivity, often with little effect on its voltage-dependent gating (*Beyder et al., 2012a*; *Beyder et al., 2012b*). Despite this abundant phenomenological evidence, it is unclear whether mechanosensitivity is intrinsic to the channel or emerges through interactions with other factors, and the mechanism of mechanosensitivity in Na$_V$ channels remains unknown.

Na$_V$ channels operate through a complex gating mechanism, where the voltage-dependent movement of the four voltage sensors can trigger a voltage-independent physical opening of the intracellular gate in the pore, immediately followed by a fast and thorough inactivation (*Patlak, 1991*). Whether applied by fluid shear stress or membrane stretch, mechanical force alters the overall voltage sensitivity of macroscopic Na$_V$ currents (*Beyder et al., 2010*; *Morris and Juranka, 2007*; *Strege et al., 2003*), but we do not know how each gating transition is influenced by force. In principle, this information could be extracted by analyzing the response of single-channel events or macroscopic currents to mechanical stimuli, as recently shown for K$_V$ channels (*Schmidt et al., 2012*). However, the complexities of the eukaryotic Na$_V$ channel structure, together with its fast activation and inactivation kinetics, would make this mechanistic analysis more challenging.

An alternative strategy is to use bacterial voltage-gated sodium channels, which have emerged as powerful models for eukaryotic Na$_V$s (*Bagnéris et al., 2014*). Like their eukaryotic counterparts, prokaryotic Na$_V$s are strongly voltage-sensitive (*Ren et al., 2001*), have similar pharmacological sensitivities (*Lee et al., 2012a*; *Lee et al., 2012b*), and share some structural elements despite being homotetramers (*Bagnéris et al., 2014*; *Catterall and Zheng, 2015*; *Lee et al., 2012b*). NaChBac from *B. halodurans* is the first prokaryotic Na$_V$ channel discovered (*Ren et al., 2001*) and presents significant advantages for mechanistic studies: at one-fourth the coding sequence length of eukaryotic Na$_V$s, NaChBac has simpler mutagenesis, structural symmetry, and thus potentially simpler gating, slower kinetics, and removable inactivation, which altogether facilitate detailed mechanistic investigations (*Lee et al., 2012a*; *Lee et al., 2012b*). In this study, we examined the mechanism of NaChBac mechanosensitivity through a combination of macroscopic and single-channel recordings, kinetic modeling, structural analysis, and mutagenesis, and found that mechanosensitivity is intrinsic and likely resides with the channel pore.

## Results

### Mechanical stimulation of bacterial voltage-gated sodium channels

We first tested if prokaryotic sodium channels are mechanically sensitive, as previously shown for eukaryotic Na$_V$s (*Beyder et al., 2010*; *Morris and Juranka, 2007*; *Strege et al., 2003*; *Figure 1*). In a side-by-side comparison with the eukaryotic Na$_V$1.5, we examined two prokaryotic channels: the wild-type (WT) NaChBac and a mutant (T220A) NaChBac with inactivation removed (*Lee et al., 2012a*; *Lee et al., 2012b*; *Figure 1A*). We expressed each channel in HEK293 cells and assayed its

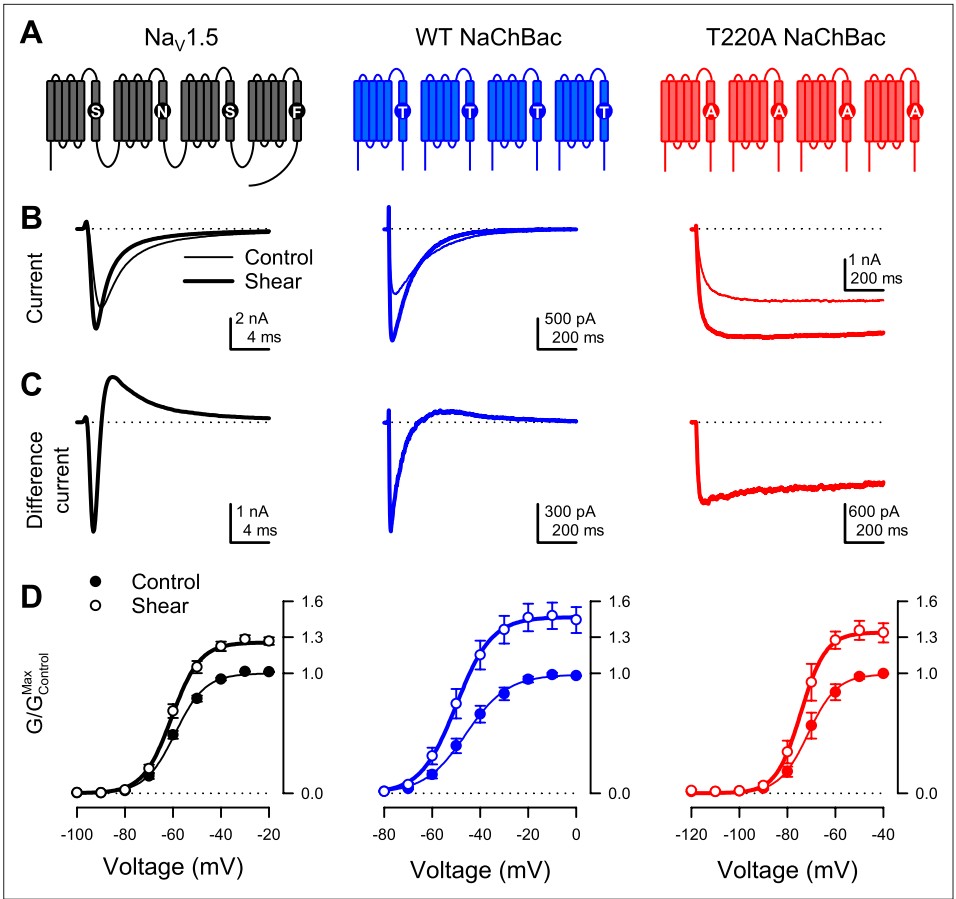

**Figure 1.** Shear stress increases the peak Na$^+$ current of eukaryotic Na$_V$1.5 and prokaryotic Na$_V$ channel NaChBac. (**A**) Topologies of eukaryotic Na$_V$ channel Na$_V$1.5 (black) and prokaryotic Na$_V$ channel NaChBac, without (WT, blue) or with (T220A, red) point mutation T220A, which makes NaChBac devoid of inactivation. (**B**) Representative Na$^+$ currents were elicited by a depolarization from –120 mV to –40 mV of Na$_V$1.5 (black), WT NaChBac (blue), or T220A NaChBac (red), before (—) or during (■) shear stress. (**C**) Difference currents were obtained by subtracting the control trace from the shear trace in (**B**). (**D**) Voltage-dependent conductance normalized to the maximum conductance of controls (G/G$_{Max,Control}$) for Na$_V$1.5 (black), WT NaChBac (blue) or T220A NaChBac (red), before (—) or during (■) shear stress (n=7–10 cells; p<0.05 by a paired two-tailed t-test when comparing shear to control at voltages >−70 mV for Na$_V$1.5, >−60 mV for WT and >−80 mV for T220A).

The online version of this article includes the following source data and figure supplement(s) for figure 1:

**Source data 1.** Whole cell conductance.

**Source data 2.** Whole cell shear stress parameters.

**Figure supplement 1.** Shear stress increases peak Na$^+$ current, hyperpolarizes the voltage of half-activation, and accelerates the kinetics of eukaryotic and prokaryotic Na$_V$ channels in HEK293 cells.

---

mechanosensitivity via whole-cell electrophysiology, with fluid shear stress (~1.1 dyn/cm$^2$) applied as mechanical stimulation. Under control conditions, the wild-type NaChBac responded to depolarizing voltage pulses with steep activation followed by complete inactivation, like Na$_V$1.5 but with slower kinetics (*Figure 1B*, *Figure 1—figure supplement 1A-D*). The T220A mutant activated and stayed open with minimal inactivation (*Figure 1B*; *Figure 1—figure supplement 1B*).

Shear stress increased the whole-cell currents of both prokaryotic channels, comparably to Na$_V$1.5 (*Figure 1B*, 'control' vs. 'shear'; *Figure 1—figure supplement 1B, E*; I$_{Peak}$ in *Table 1*). Both activation and inactivation responded to shear stress, as demonstrated by the difference currents (I$_{Shear}$ – I$_{Control}$) from both wild-type NaChBac and Na$_V$1.5 (*Figure 1C*). Removal of inactivation in NaChBac T220A allowed us to separate these responses and focus on activation. Shear forces also increased T220A NaChBac currents, albeit slightly less than for wild-type (*Figure 1C*), suggesting that mechanical

**Table 1.** Effect of shear stress on parameters of wild-type and T220A NaChBac.

| | Nav1.5 | | | WT NaChBac | | | T220A NaChBac | | |
|---|---|---|---|---|---|---|---|---|---|
| | **Control** | **Shear** | **Change** | **Control** | **Shear** | **Change** | **Control** | **Shear** | **Change** |
| $I_{PEAK}$ (pA/pF) | -134.3±16.4 | -164.0±18.5* | +23.6 ± 3.5% | -37.0±9.1 | -59.2±15.5* | +58.7 ± 10.1% | -214.6±60.4 | -281.8±73.7* | +39.0 ± 6.8% |
| $G_{MAX}$ (nS) | 2.21±0.28 | 2.75±0.32* | +26.2 ± 3.2% | 0.48±0.09 | 0.71±0.15* | +47.0 ± 10.9% | 2.96±0.81 | 3.72±0.95* | +31.7 ± 8.3% |
| $E_{REV}$ (mV) | +23.9 ± 2.3 | +20.1 ± 2.2* | -3.8±0.4 | +55.6 ± 5.9 | +55.2 ± 5.3 | -0.3±2.4 | +21.9 ± 2.4 | +18.8 ± 2.5 | -3.1±1.7 |
| $V_{1/2A}$ (mV) | -59.1±0.8 | -60.5±1.0 | -1.4±0.6 | -45.1±2.5 | -49.6±2.1* | -4.4±0.6 | -70.8±2.3 | -74.5±2.2* | -3.7±0.9 |
| $V_{1/2I}$ (mV) | -93.0±2.1 | -95.5±2.4* | -2.4±0.4 | -56.9±2.8 | -60.7±2.0* | -3.7±1.1 | -44.1±5.4 | -56.4±3.5* | -12.2±3.1 |
| $\delta V_A$ | 6.1±0.3 | 5.7±0.3* | -0.4±0.1 | 8.1±0.6 | 6.8±0.3* | -1.3±0.4 | 5.1±0.6 | 3.2±0.6 | -1.9±0.8 |
| $\delta V_I$ | -6.9±0.1 | -6.7±0.1* | 0.2±0.1 | -6.0±0.2 | -5.8±0.3 | 0.2±0.3 | -14.3±1.9 | -13.2±2.3 | 0.4±2.2 |
| $\tau_A$ (ms) | 0.49±0.04 | 0.43±0.03* | -10.5 ± 6.0% | 18.6±3.4 | 11.6±2.5* | -39.3 ± 3.8% | 8.4±1.8 | 4.5±0.7* | -42.1 ± 5.6% |
| $\tau_I$ (ms) | 0.77±0.07 | 0.53±0.04* | -29.8 ± 3.4% | 213.0±37.8 | 162.4±31.6* | -23.3 ± 4.3% | — | — | — |

Shear, the flow of extracellular solution; $I_{Peak}$, maximum peak current density; $G_{Max}$, maximum peak conductance; $E_{Rev}$, reversal potential; $V_{1/2a}$, half-point of steady-state activation; $V_{1/2i}$, half-point of steady-state inactivation; $\tau_a$, time constant of activation at -30 mV; $\tau_i$, time constant of inactivation at -30 mV; $\delta V_a$, slope of steady-state activation; $\delta V_i$, slope of steady-state inactivation; The background of Nav1.5 was H558/Q1077del. Number of cells: Nav1.5, 10; wild-type (WT) NaChBac, 7; T220A NaChBac, 7.

*p<0.05 shear vs. control by a two-tailed paired Student's t-test.

forces act predominantly on the mechanistic steps associated with the channel's activation and/or opening. Overall, shear stress increased maximum conductance ($G_{Max}$) by 47% for WT NaChBac and 34% for T220A NaChBac, compared to 26% for $Na_V1.5$ (*Figure 1D*, $G_{Max}$ in *Table 1*).

Although the steady-state conductance curves obtained under shear stress mostly appear as vertically stretched versions of the control curves, accounting for the higher maximum current, they exhibit a slight negative shift of the half-activation voltage (*Figure 1D*; $V_{1/2a}$ in *Table 1*). This effect is more easily visualized when each conductance curve is normalized to its maximum (*Figure 1—figure supplement 1F*). Shear stress also increased the conductance slope ($\delta V_a$ in *Table 1*). Interestingly, the half-inactivation voltage also exhibits a negative shift (*Figure 1—figure supplement 1G*; $V_{1/2i}$ in *Table 1*). Kinetically, shear stress accelerates the time course of both activation (*Figure 1—figure supplement 1C*; $\tau_a$ in *Table 1*) and inactivation (*Figure 1—figure supplement 1D*; $\tau_i$ in *Table 1*).

## Interactions between electrical and mechanical stimuli

The whole-cell shear stress experiments demonstrate that mechanical forces affect NaChBac macroscopic currents. These results are likely to have mechanistic implications, but ambiguities inherent

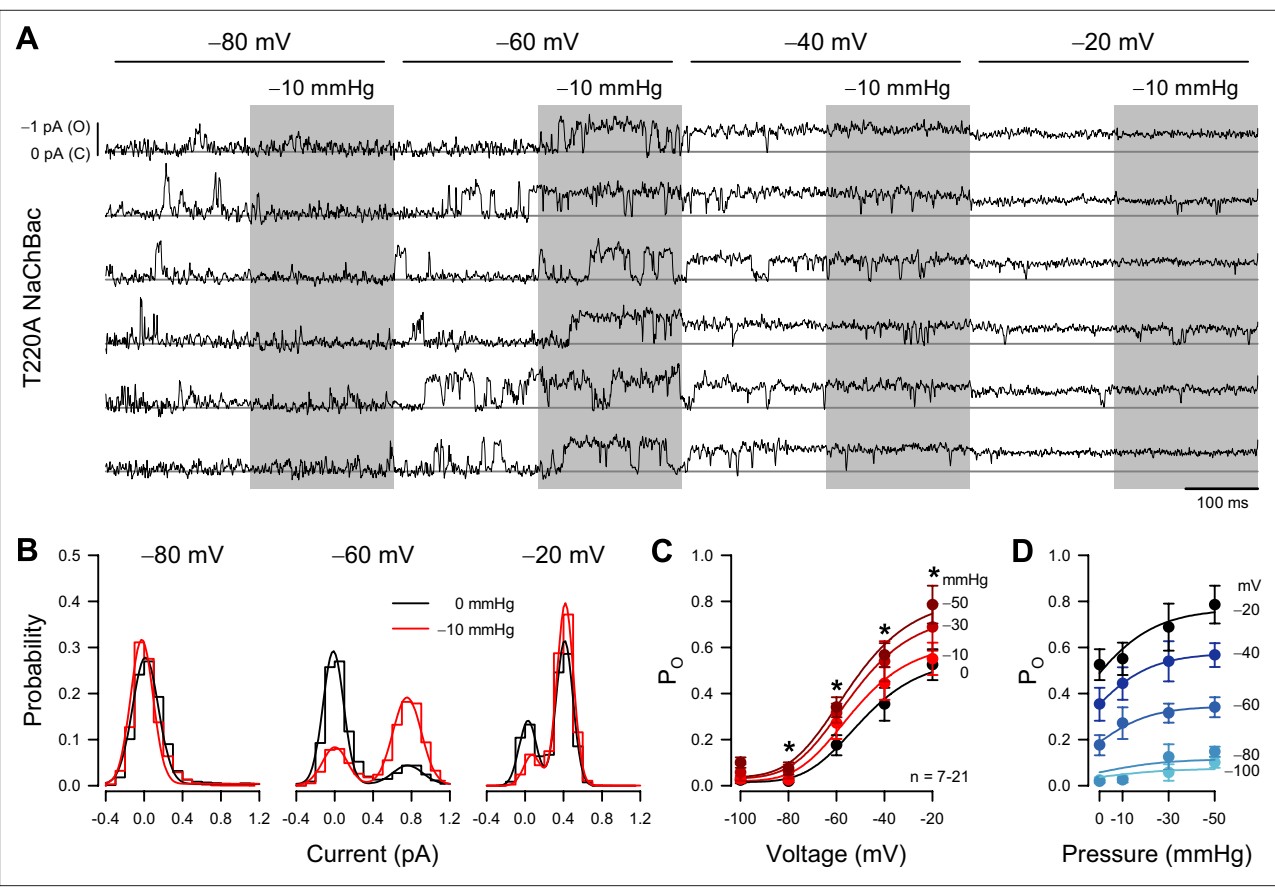

**Figure 2.** Patch pressure increases the open channel probability of T220A NaChBac single channels in P1KO cells. (**A**) Representative traces of single T220A NaChBac channels at −80, −60, −40, or −20 mV and with 0 (unshaded) or −10 mmHg (shaded region) applied to the patch. (**B**) All-point histograms constructed from the traces shown in (**A**) at −80, −60, or −20 mV and 0 (black) or −10 mmHg (red) binned every 0.2 pA. Bins were normalized to an area of 1 and fit with a sum of two Gaussians, in which open events at −60 mV were 0.77 pA and 0.17 $P_O$ without pressure and 0.75 pA and 0.72 $P_O$ (330% increase) with pressure; open events at −20 mV were 0.43 pA and 0.90 $P_O$ without pressure and 0.42 pA and 0.90 $P_O$ (0% increase) with pressure. (**C**) Mean open probabilities ($P_O$) at voltage steps from −100 to −20 mV with 0 (black) or −10 to −50 mmHg (red gradient) pressure (n=7–21 cells per voltage; *p<0.05, control vs. pressure by a paired two-tailed t-test). (**D**) $P_O$ per voltage from (**C**), re-plotted vs. pressure (0 to −50 mmHg).

The online version of this article includes the following source data and figure supplement(s) for figure 2:

**Source data 1.** Single channel open probability.

**Source data 2.** Endogenous single channel activity.

**Figure supplement 1.** Endogenous channels in Piezo1-KO HEK (P1KO) cells are insensitive to pressure stimulus.

to macroscopic currents limit the information that can be extracted from data about individual state transitions. We addressed these ambiguities via single-channel recordings, followed by a mechanistic analysis to determine how force interacts with voltage to gate the channel. To simplify experiments and interpretations, we focused on NaChBac T220A, which lacks inactivation (*Lee et al., 2012a*; *Lee et al., 2012b*). We expressed NaChBac T220A in Piezo1-knockout (P1KO) HEK293 cells, free of mechanosensitive channel activity (*Dubin et al., 2017*; *Figure 2A*, *Figure 2—figure supplement 1A-F*). We assayed mechanosensitivity via cell-attached patch-clamp electrophysiology, using a high-speed pressure clamp (*Besch et al., 2002*) to apply controlled suction to patches.

The single-channel amplitude of voltage-gated sodium channels is tiny (~1 pA at –80 mV and ~0.5 pA at –20 mV), and pressure-clamping introduces additional noise and transient artifacts. Together with rapid channel kinetics, these limitations have traditionally prevented single-channel studies on mechanosensitivity in VGICs. After careful mechanical and electrical optimization, despite the low signal-to-noise ratio typical for sodium channels (*Vandenberg and Bezanilla, 1991*), and the noise introduced by the pressure clamp (*Figure 2—figure supplement 1G*), we were able to resolve single-channel events across a physiologically relevant voltage range, and with enough bandwidth (~1 kHz) to capture sufficiently fast kinetics (*Figure 2A*).

Suction on the membrane patch exerts a mechanical force on the channel (*Coste et al., 2010*). Because patches have non-zero resting tension (*Suchyna et al., 2009*), we designed stimulation protocols to test voltage- and mechano-sensitivity in a pairwise fashion (*Figure 2A*), enabling us to assess mechanosensitivity from the difference between the suction-induced currents and the no-suction baseline, for all channels and traces. Under these conditions, a non-zero patch tension is expected to slightly bias the kinetic properties at rest but not obscure the magnitude and location of mechanosensitive steps within the gating mechanism. Within each 400ms voltage step from –100 to –20 mV, the suction pressure alternated between 0 and –10, –30, or –50 mmHg. Thus, we could obtain and compare control and pressure data in the same cell, using test pressures relevant to mechanosensitive channel function (*Coste et al., 2010*; *Gottlieb et al., 2012*). As indicated by the current amplitude histograms (*Figure 2B*), the single-channel current is less than 0.5 pA at –20 mV, but we could still separate the closed and open levels. Above –20 mV, the unitary current became too small for reliable analysis. Using a half-amplitude threshold method, we measured open-state occupancy between –100 and –20 mV (*Figure 2C*). We cross-checked this approach against fitting all-point amplitude histograms with sums of two Gaussian distributions, one for each current level (*Figure 2B*), where the relative weight of the open-level Gaussian indicates the open-state occupancy probability ($P_O$). The two methods produced similar results.

Under control conditions (zero applied patch pressure), $P_O$ was strongly voltage-dependent (*Figure 2A–C*), as predicted by the whole-cell activation curve (*Figure 1D*). $P_O$ was nominally zero at –80 mV and below, and $P_O$ increased as the voltage became more positive, reaching 0.525 at –20 mV. Relative to whole-cell activation, the $P_O$ curve is shallower and ~20 mV more positive. This discrepancy is likely an artifact of a scattered and non-zero resting potential, unmeasurable in cell-attached recordings (averaging sigmoid curves with a scattered and shifted midpoint results in a shallower and shifted sigmoid).

Patch suction altered the voltage-dependent $P_O$ (*Figure 2A–C*; *Table 2*). At extremely negative voltages (–100 and –80 mV), where the channel is closed under control conditions, $P_O$ remained zero under suction. However, pressure significantly increased $P_O$ at more positive voltages. Responses were dependent on suction strength (*Figure 2C and D*), but even at high negative pressures (–30 and –50 mmHg), the induced changes were confined to the voltage activation range (–80 to –20 mV) (*Figure 2C and D*). These results agree with the whole-cell experiments, where shear stress stretched the curve vertically. As single-channel data yield the actual $P_O$ values under different pressures and voltages, we could establish that the increase in whole-cell conductance results from an increase in $P_O$ and not in single-channel conductance, which remained constant under pressure (*Figure 2A and B*).

Because some previous studies have shown that shear stress and patch pressure can create irreversible changes (*Beyder et al., 2010*; *Schmidt et al., 2012*; *Wang et al., 2009*), we tested specifically for reversibility in our preparations. In whole-cell experiments, we found that the increase in peak Na$_V$1.5 and NaChBac T220A current density induced by shear stress are fully reversible (*Figure 3A–B*, *Figure 3—figure supplement 2*), although in some cells the acceleration in Na$_V$1.5 kinetics or shift in half-activation voltage was not reversible and led to a non-zero difference current (*Strege et al.,*

**Table 2.** Effect of pressure on the open probability of mutants D93A and I228G in the T220A NaChBac background.

| Voltage | T220A background | | | D93A | | | I228G | | |
|---|---|---|---|---|---|---|---|---|---|
| (mV) | Control | Pressure | Difference | Control | Pressure | Difference | Control | Pressure | Difference |
| −100 | 0.023±0.013 | 0.028±0.014 | 0.004±0.002 | 0.079±0.022 | 0.109±0.062 | 0.030±0.043 | 0.021±0.009 | 0.019±0.008 | −0.002±0.001 |
| −80 | 0.019±0.005 | 0.024±0.009 | 0.005±0.005 | 0.135±0.023 | 0.237±0.048* | 0.103±0.037† | 0.028±0.020 | 0.032±0.019 | 0.003±0.002 |
| −60 | 0.176±0.044 | 0.271±0.069 | 0.096±0.043 | 0.471±0.082 | 0.554±0.080* | 0.082±0.014 | 0.100±0.033 | 0.114±0.036 | 0.014±0.011† |
| −40 | 0.353±0.071 | 0.443±0.070* | 0.090±0.025 | 0.657±0.051 | 0.665±0.045 | 0.008±0.023† | 0.379±0.062 | 0.391±0.066 | 0.012±0.011† |
| −20 | 0.525±0.067 | 0.551±0.070* | 0.026±0.010 | 0.638±0.011 | 0.611±0.015 | −0.027±0.016† | 0.537±0.069 | 0.524±0.067 | −0.012±0.010 |

Open probability; n = 6–12 cells.

*p<0.05, −10 vs. 0 mmHg pressure, by a two-tailed paired t-test.

†p<0.05, D93A or I228G vs. T220A background by a two-tailed unpaired t-test.

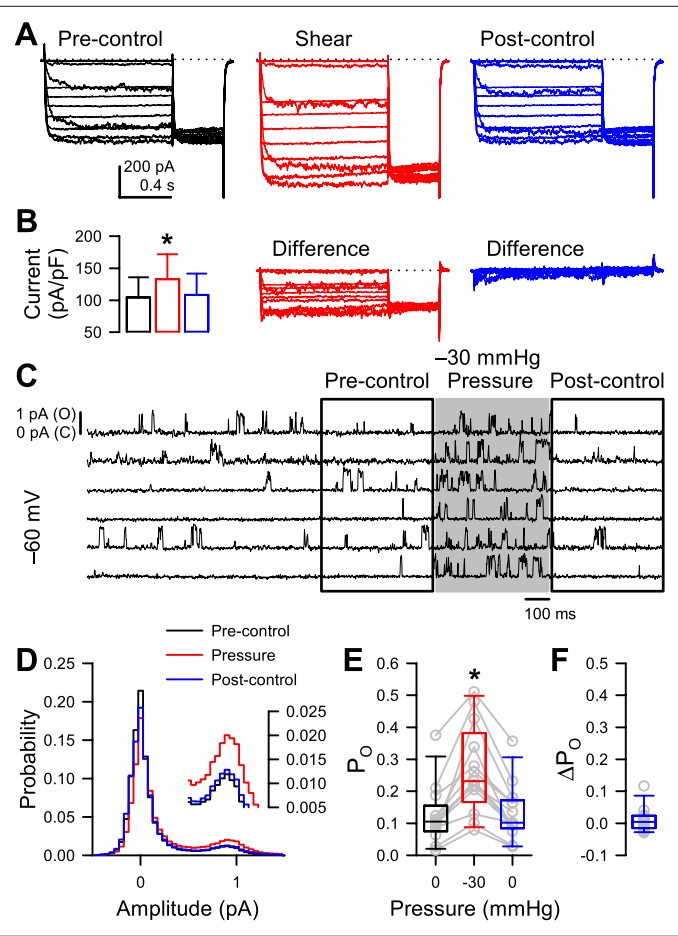

**Figure 3.** Mechano-sensitive increase in whole-cell peak currents and single-channel open probability of T220A NaChBac is reversible. (**A**) Representative whole-cell currents from HEK cells expressing T220A NaChBac were elicited by a voltage protocol (*Figure 1—figure supplement 1A*) before (black), during (red), or after (blue) shear stress. (**B**) Peak current densities before (black), during (red), or after (blue) shear stress (n=5 cells, *p<0.05 to pre-control by a one-way ANOVA with Dunnett's post-test). (**C**) Representative single channel activity at –60 mV from Piezo1-knockout HEK cells transfected with T220A NaChBac, before (unshaded), during (shaded region), or after application of –30 mmHg to the patch for 500 ms. (**D**) All-sample distributions of single channel activity from the cell shown in (**C**), binned every 0.05 pA with peaks at 0 pA (closed) and ~0.9 pA (open). (**E**) Mean open channel probability ($P_O$) per cell (gray circles) before (black), during (red), or after (blue) application of –30 mmHg pressure. (**F**) Differences in post-pressure $P_O$ ($\Delta P_O$) from pre-pressure controls.

The online version of this article includes the following source data and figure supplement(s) for figure 3:

**Source data 1.** Reversibility.

**Source data 2.** Single channel reversibility.

**Source data 3.** Whole cell reversibility.

**Figure supplement 1.** Effect of pressure on voltage-dependent open probability.

**Figure supplement 2.** Shear-sensitive increase in whole-cell peak currents of $Na_V1.5$ is reversible.

---

*2003*; *Figure 3—figure supplement 2B*). With single channels, to test the reversibility of $P_O$ increase by patch pressure, we lengthened the time before pressure application to 2 s, applied –30 mmHg pressure for 500ms, and compared the pre- and post-pressure $P_O$ values (*Figure 3C*, *Figure 3—figure supplement 1A*). Pressure increased $P_O$ throughout the –80 to –20 mV activation range (*Figure 3—figure supplement 1B*), with 20 out of 21 cells responding at –60 mV (*Figure 3D–E*). Once pressure returned to 0 mmHg, $P_O$ returned to its baseline value (*Figure 3F*, *Figure 3—figure supplement 1C-D*). As expected, this change was not instantaneous, because the channel must transition back into a different set of state occupancies, which takes time (*Figure 3—figure supplement 1B*).

## Mechanical force mainly affects pore opening

An intuitive interpretation of the whole-cell and single-channel results is that force alone does not open the channel. If it did, we would see openings at voltages where the channel is typically closed, provided that we applied enough membrane tension. Instead, we see that force enhances openings (increases $P_O$) that are already driven by membrane depolarization. A simple interpretation is that force does not create additional conformational states but modifies the energetics of the existing transitions. If this is true, then force will interact with at least one mechanistic component: (1) voltage sensor activation, (2) pore opening, or (3) inactivation. It seems to us that inactivation is unlikely to play a significant role. First, NaChBac T220A responds to patch pressure like the wild type does, even though the mutant virtually lacks inactivation (*Figure 1B and C*). Second, eukaryotic $Na_V$ and wild-type NaChBac have similar responses to shear stress (*Figure 1B and C*), even though they inactivate via different mechanisms (*Gamal El-Din et al., 2019*). Thus, the effects of force on inactivation could simply be due to the coupling of inactivation to activation (*Aldrich et al., 1983*). For these reasons, we focus here on the NaChBac T220A channels, which show minimal inactivation.

The remaining possibilities are that force interacts with (1) the voltage sensors or (2) the pore. While not necessarily mutually exclusive, the two extreme models corresponding to these interactions are easier to formulate and discriminate than mixed models. Hence, we examined the specific changes in kinetic properties driven by force and compared them against model predictions. We first formulated a kinetic model (*Figure 4A*) that encapsulates the homo-tetrameric nature of NaChBac T220A, its voltage-dependent activation, and its lack of inactivation. We made the rates along the activation pathway (closed states $C_1$ to $C_5$) strongly voltage-dependent to agree with the whole-cell and single-channel activation curves (*Figures 1D and 2C*). In contrast, we made the concerted opening transition ($C_5$ to open state $O_6$) voltage-independent, as previously shown for eukaryotic $Na_V$s (*Kuo and Bean, 1994*) and based on our observation that the whole-cell activation curve reaches a steady maximum (*Figure 1D*), which, according to the single-channel data, corresponds to a maximum $P_O$ of ~0.6 (*Figure 2C*). If the concerted opening were significantly voltage-dependent, the maximum $P_O$ would approach unity at strongly depolarizing voltages. The model parameters were manually adjusted to match the experimental data under control conditions (see Methods).

## Mechanosensitive activation

The first scenario, where mechanical force interacts only with the voltage sensors, is captured by a mechanosensitive activation (MSA) model (*Figure 4A*). In this case, we expect to see force-induced changes in the mechanosensitive rate constants along the $C_1$ to $C_5$ pathway. Experimentally, we observed increased whole-cell current by shear stress (*Figure 4B*), matched by an increase in $P_O$ when membrane tension is raised via patch suction (*Figure 4C*). With the MSA model, we can explain this result by ascribing positive tension sensitivity (i.e. negative pressure sensitivity) to the activation (forward) rates and/or negative tension sensitivity to the deactivation (backward) rates. A situation where both activation and deactivation rates have positive or negative tension sensitivities is also acceptable, as long as the forward sensitivities are more positive than the backward ones.

The MSA model predicts that the activation curve shifts toward more negative voltages when tension increases, but its slope and maximum value remain precisely the same (*Figure 4B*, MSA). The activation midpoint would change because tension shifts the equilibrium of each activation step ($C_1$ to $C_5$) toward $C_5$ at any given voltage. In contrast, the slope and maximum $P_O$ would be unchanged by tension because they are determined by the voltage sensitivity of activation and by the voltage- and force-independent opening transition ($C_5$ to $O_6$), respectively. In other words, extreme tension would push the channel to reside in the $C_5$ and $O_6$ states, but the equilibrium between these two states – and hence maximum $P_O$ – would remain the same. However, we did not observe this behavior experimentally. Instead, when membrane tension increased, both the whole-cell activation curve (*Figure 4B*) and the $P_O$ curve (*Figure 4C*) exhibited increased steepness and greater maximum value. The experimental activation data are thus in stark contrast with the predictions of the MSA model.

## Mechanosensitive opening

The alternative scenario, where mechanical force interacts only with the channel pore, is captured by a mechanosensitive opening (MSO) model (*Figure 4A*). In this case, we expect to see force-induced changes in the mechanosensitive $C_5$ to $O_6$ rate constants. With the MSO model, the observed increase

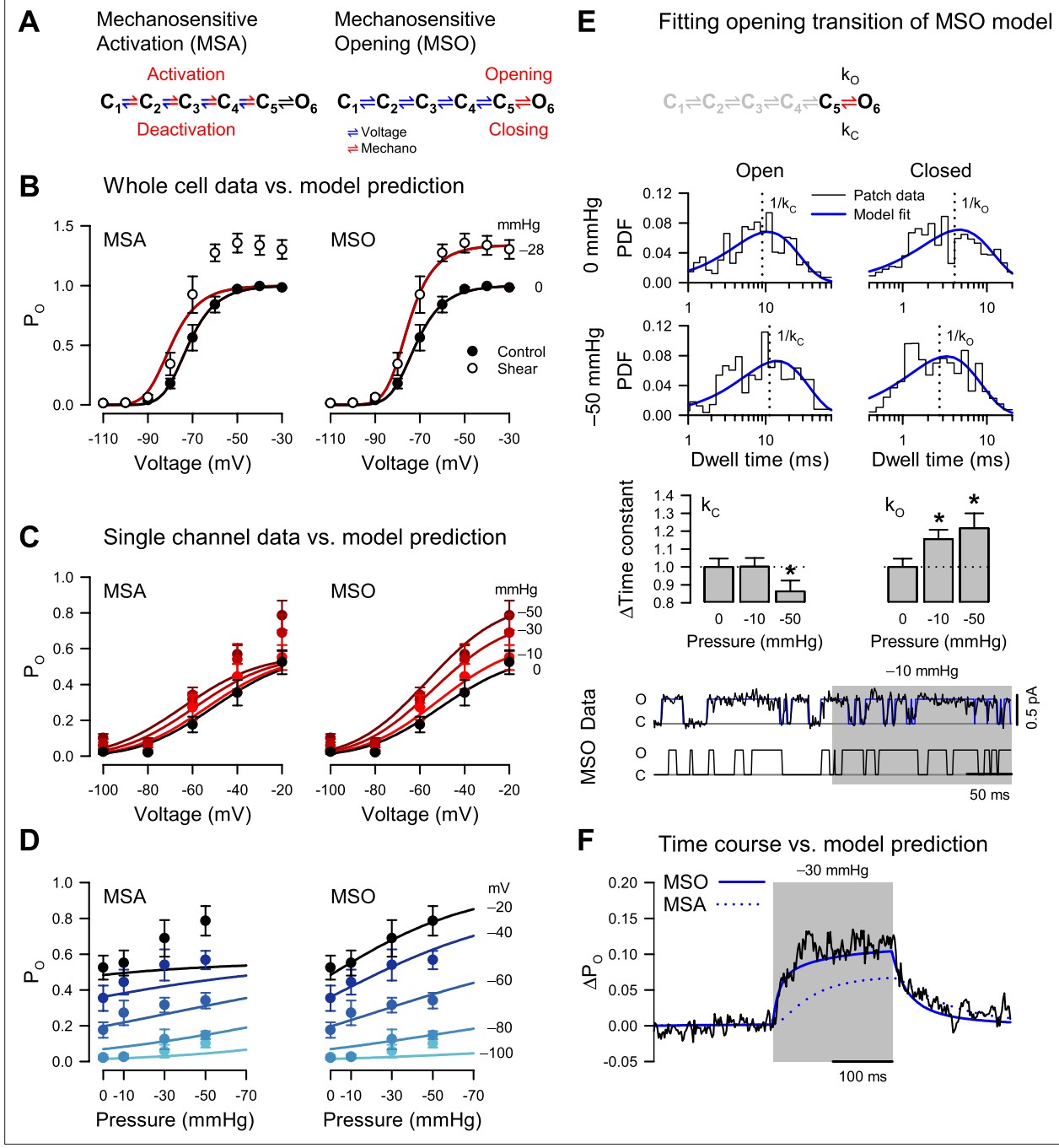

**Figure 4.** Pressure destabilizes the T220A NaChBac closed state. (**A**) Mechanosensitive activation (MSA) depicts a model in which the $C_1$ to $C_5$ closed state transitions are both voltage- and pressure-dependent (blue and red); mechanosensitive opening (MSO) depicts a model in which the $C_1$ to $C_5$ closed state transitions are voltage-dependent (blue), and the $C_5$ closed to $O_6$ open state transition is pressure-dependent (red). The predictions of these two models to voltage and pressure stimuli are shown in (**B–D**), with kinetic parameters as described in Materials and Methods. (**B**) MSA (left) and MSO (right) model predictions of open probability ($P_O$) across voltages from –110 to –30 mV with 0 (black) or –28 mmHg applied pressure (dark red), compared to $G/G_{Max}$ whole-cell data (*Figure 1D*) with 0 (●) or 10 mL/min (○) fluid shear stress. (**C–D**) MSA (left) and MSO (right) model predictions of single channel $P_O$ (●) plotted versus voltage (**C**) at pressures from 0 to –50 mmHg (red gradient) or versus pressure (**D**) at voltages from –100 to –20 mV (blue gradient). (**E**) MSO model adapted fit to a single pressure-sensitive $C_5$ to $O_6$ transition with pressure-dependent kinetic constants assigned for opening ($k_O$) and closing ($k_C$). Insets: top, open (left), and closed (right) dwell time histograms of single channel data (black) vs. the MSO model PDF curves (blue), under 0 mmHg (top row) or –50 mmHg pressure (bottom row), with vertical dotted lines indicating the inverse of the time constants; middle, bar graphs depicting the change in the time constants $k_C$ (left) or $k_O$ (right) with –10 or –50 mmHg pressure; bottom, single channel trace recorded at –20 mV (black) and idealization (blue) with –10 mmHg applied to the region shaded (gray), compared to a trace simulated with the MSO

*Figure 4 continued on next page*

*Figure 4 continued*

model. *p<0.05 to 0 mmHg by unpaired two-tailed t-tests using the raw values of the time constants. (**F**) MSA (dotted blue line) and MSO (solid blue line) model prediction of single channel $P_O$ at –60 mV before, during, and after pressure, compared to the average current from single channel data (black).

The online version of this article includes the following source data for figure 4:

**Source data 1.** Modeling.

in $P_O$ by tension can be explained by ascribing positive tension sensitivity to the opening (forward) rate, and/or negative tension sensitivity to the closing (backward) rate, or any combination where the forward sensitivity is more positive than the backward one.

The MSO model predicts that the activation curve reaches a larger value and becomes steeper when tension increases and shifts slightly toward more negative voltages (*Figure 4B*, MSO). The maximum $P_O$ would change because it is determined by the tension-dependent pore opening rates, but why would the voltage activation curve shift and steepen under tension, when the tension-dependent rates are voltage-insensitive? The reason is that voltage acts through the voltage-dependent activation/deactivation rates to increase the joint occupancy of the final two states, $C_5$ and $O_6$, while tension acts through the tension-dependent opening rates to increase the occupancy of the open state $O_6$. Thus, under tension, an increase in voltage will lead to a proportionately larger increase in $P_O$, compared to zero-tension conditions, and cause a shift in the activation curve, increased steepness, and a greater maximum value. Indeed, the MSO model supports the mechanically-induced changes in the whole-cell and single-channel activation curves (*Figure 4B and C*, MSO).

Having examined the changes in $P_O$ vs. voltage under different pressure values, we conversely examined $P_O$ vs. tension under different voltages (*Figure 4D*). Reversing voltage and tension as independent variables does not create new information, as we are using the same data points as in *Figure 4C*, but it makes it easier to judge the fitness of each model. Thus, the MSA model predicts a significant shift in the $P_O$ vs. tension curve when the voltage increases but no change in the maximum value and the slope of the curve (*Figure 4D*, MSA). In contrast, the MSO model predicts a significant change in the maximum value and the slope but only a small shift in the curve (*Figure 4D*, MSO). The experimental $P_O$ data points align well with either the MSA or the MSO model at zero pressure. However, the MSO model becomes a significantly better match to the data as the pressure increases (*Figure 4C*).

## Mechanical force destabilizes the NaChBac closed state

The analysis so far clearly favors the MSO model. However, we used only the steady-state information in the data, and we do not know if the MSO model can also explain the observed kinetics. The MSO model assumes tension-dependent opening and closing rates (at least one, if not both), whereas the MSA model assumes these rates to be tension-independent. If the pore opening transition were tension-dependent, then the pore opening ($C_5$ to $O_6$) and/or the closing ($O_6$ to $C_5$) rate would be affected by force, which would be reflected in the single-channel closed and open lifetimes. In our simple NaChBac kinetic model, the open state lifetime distribution has only one component, with the time constant equal to the inverse of the closing rate constant ($O_6$ to $C_5$). In contrast, the closed-state lifetime distribution has five components, without an easy way to isolate the opening rate constant. However, the deactivation rates are likely so small at extremely depolarizing voltages (e.g. ≥–20 mV) that the channel essentially flickers between the last two states ($C_5$ and $O_6$). Hence, as an approximation, the closed lifetime distribution has only one component at these extreme voltages, with a time constant that approaches the inverse of the opening rate constant ($C_5$ to $O_6$). Consequently, a truncated model with only the final two states would approximate the channel at –20 mV (*Figure 4E*).

Because NaChBac T220A has some residual inactivation (*Figure 1—figure supplement 1E, J*), we used relatively short (200–500 ms) voltage/pressure stimulation episodes, so many recorded traces contained no events. To fit the single-channel data with the MIL algorithm (*Qin et al., 1996*), we had to discard the first and last dwells in each trace because they are by necessity truncated and cannot be used for analysis, which means that all the eventless traces were also discarded. Under these conditions, the remaining traces that are suitable for analysis would slightly bias the estimated rates because of the inherently higher $P_O$. Nevertheless, the mechanosensitivity of the opening and closing rates should emerge clearly from this analysis. As a verification, we also performed the analysis

with the model parameters constrained (*Navarro et al., 2018*; *Salari et al., 2018*) to enforce a ratio between the opening and closing rate constants corresponding to the $P_O$ measured under control (zero added tension) conditions, and also to enforce the total pressure sensitivity, which can be reliably estimated from the $P_O$ data. The results obtained with these parameter constraints were similar to those obtained in the constraint-free analysis.

The closed state lifetime distribution shifts toward shorter dwell times by 15% under –10 mmHg pressure ($k_O$: 124.9 ± 5.7 s$^{-1}$ at 0 mmHg to 144.4 ± 6.6 s$^{-1}$ at –10 mmHg; n=124 traces from 10 patches) and by 21% under –50 mmHg pressure ($k_O$: 178.2 ± 11.9 s$^{-1}$ at 0 mmHg to 217.0 ± 14.7 s$^{-1}$ at –50 mmHg; n=23 traces from three patches) (*Figure 4E*). The average closed lifetime approaches the bandwidth limit (~1 ms) and, even though the fitting algorithm partially compensates for the missed events, it's possible that the increase in the opening rate with pressure is underestimated. In contrast, the open state distribution remained virtually unchanged by tension under –10 mmHg pressure ($k_C$: 48.1 ± 2.2 s$^{-1}$ to –48.2 ± 2.3 s$^{-1}$), although it shifts toward longer dwell times under –50 mmHg ($k_C$: 101.4 ± 6.8 s$^{-1}$ to –87.5 ± 6.1 s$^{-1}$).

The observed shift in the closed state lifetimes further confirms that the channel is better represented by the MSO model, as the competing MSA model would exhibit no such shift at saturating voltages. Moreover, it suggests that force destabilizes the closed state, as the opening rate changes the most with tension. As we now have an idea about the magnitude of opening and closing dwell times, we can also examine activation kinetics. In principle, we can extract this information by fitting the single-channel data recorded at intermediate voltages (e.g. –60 mV), where the channel visits all states. However, the changes in voltage and pressure stimuli make these data non-stationary, and a more straightforward approach is to examine the macroscopic data created by averaging the single-channel recordings. As shown in *Figure 4F*, the MSO model captures well the time course of the average current and gives us an idea about the magnitude of the activation rates. In all, our modeling of the whole-cell and single channel results suggest that the MSO model, which assigns tension sensitivity to the voltage-insensitive pore opening step, best fits the experimental data and associates the NaChBac mechanosensor with the pore structure.

## Pressure may affect the stability of the intracellular gate

According to the 'force-from-lipid' model (*Martinac et al., 1990*), ion channels gain mechanosensitivity when their cross-section expands or shrinks upon a conformational change (*Perozo et al., 2002a*; *Sachs and Morris, 1998*). Based on our kinetic analysis, the site of mechanosensitivity in NaChBac is most likely the pore opening, the final gating transition ($C_5$ to $O_6$ in the MSO model in *Figure 4A*). Interestingly, previous structural modeling studies have predicted that when voltage sensors are suitably activated, mechanical energy is required to open the gate (*Fowler and Sansom, 2013*), which implies that negative membrane tension (i.e. patch suction) would facilitate opening. If our hypothesis were true, we would predict a change in the cross-section between the final two states in the MSO model: the activated but still closed $C_5$ and the open $O_6$. To test this hypothesis, we examined the two existing prokaryotic voltage-gated sodium channel structural models: Na$_V$Ab, capturing the channel in the closed conformation (*Boiteux et al., 2014*), and Na$_V$Ms, representing the open state (*McCusker et al., 2012*).

By contrasting closed and open models, we searched for the channel substructures undergoing the largest movements within the membrane plane and found that the intracellular portion of the pore-forming S6 segment is displaced laterally around a 'gating hinge' (*Figure 5A and B*). Interestingly, this type of movement has been previously proposed in functional studies (*Beyder and Sachs, 2009*; *Webster et al., 2004*; *Zhao et al., 2004*) and confirmed by structural experiments (*Lenaeus et al., 2017*), including an example where the intracellular side of a VGIC pore was found to expand the area of the bilayer's inner leaflet upon S6 lateral movement (*Beyder and Sachs, 2009*; *Iwasa et al., 1980*).

According to our structural analysis, the 'force-from-lipid' model applies to NaChBac. Because S6 helices move and open the pore only after voltage sensors activate, it follows that mechanosensitivity, which is associated with S6 movement, resides with the pore opening ($C_5$ to $O_6$ in the MSO model) and not with the voltage sensor activation ($C_1$ to $C_5$ in the MSA model). This interpretation agrees with the MSO model, with one potential caveat being that Na$_V$Ab as a closed channel could represent other closed states along the activation pathway, rather than the fully activated closed conformation ($C_5$ in the MSO model). As a result, a mechanosensitive transition could still occur before the pore opening.

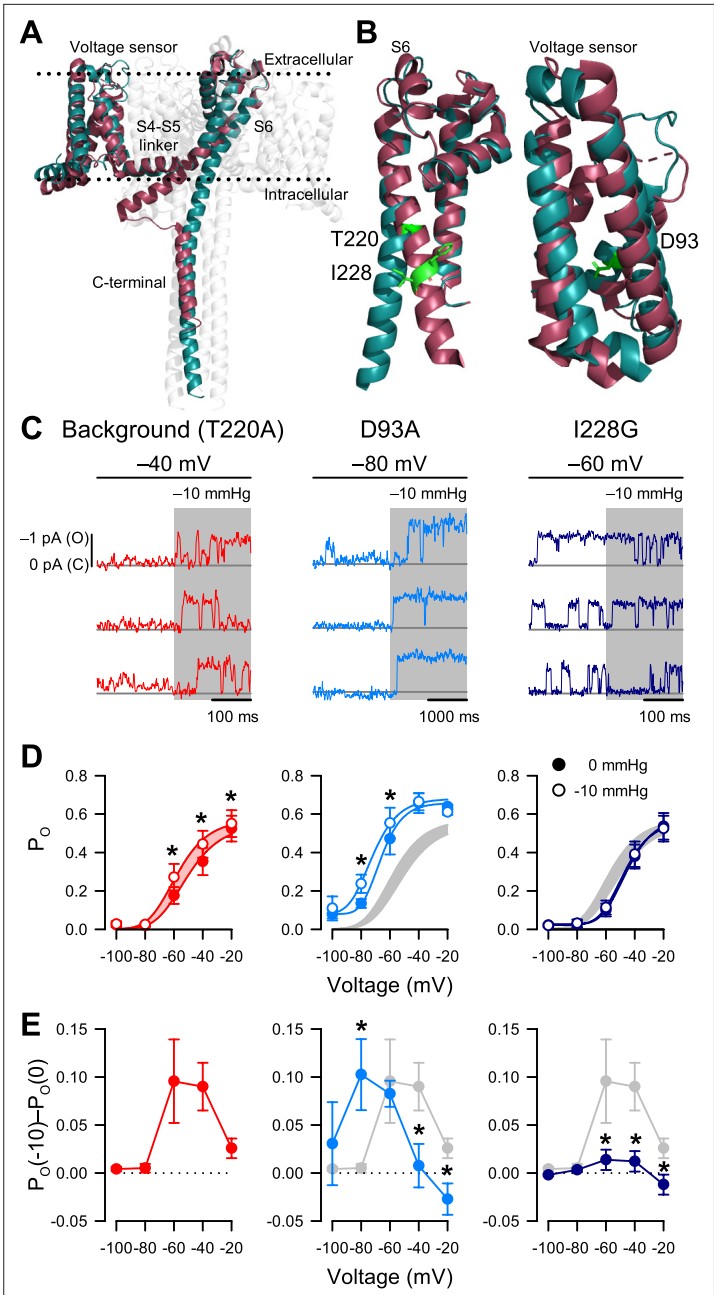

**Figure 5.** I228G disrupts the pressure sensitivity of NaChBac background T220A. (**A**) Conformational change of prokaryotic Na$^+$ channels from the closed (cyan, Na$_V$Ab, 2017) to open state (magenta, Na$_V$Ms, 2017), illustrating the movement of the voltage sensor, S4-S5 linker, S6 segment, and C-terminal tail in relation to the lipid bilayer. (**B**) Location of key residues T220A and I228 in the S6 pore segment and D93 in the voltage sensor. (**C–D**) Voltage-dependent open probabilities ((**D**), P$_O$) of single channel activities (**C**) recorded at the indicated voltages with 0 or –10 mmHg pressure from P1KO cells expressing the T220A NaChBac background (red or gray shading) or with additional mutations D93A (blue) or I228G (indigo). (*$p<0.05$, –10 mmHg vs. 0 mmHg by paired two-tailed t-tests, n=338–636 traces per voltage from 6 to 12 cells). Half-points of open probability (0 *to* –10 mmHg): T220A, –45.6 *to* –58.1 mV; D93A, –65.1 *to* –72.3 mV; I228G, –46.2 *to* –48.0 mV. (**E**) Difference in open probability induced by –10 mmHg pressure (P$_O$(–10)–P$_O$(0)) as a function of voltage in the control background (red or gray shading) or with D93A (blue) or I228G (indigo) (*$p<0.05$, D93A or I228G to T220A background by unpaired two-tailed t-tests).

The online version of this article includes the following source data and figure supplement(s) for figure 5:

**Source data 1.** Mutant pressure sensitivity.

*Figure 5 continued on next page*

*Figure 5 continued*

**Source data 2.** Mutant voltage dependence.

**Source data 3.** Macroscopic currents.

**Figure supplement 1.** Whole-cell voltage-dependent Na$^+$ currents elicited from P1KO cells transfected with NaChBac mutants D93A or I228G in the T220A background.

**Figure supplement 2.** Effect of pressure on I228G single channel open probability and macroscopic patch currents.

In other words, although the pore opening is likely mechanosensitive, it might not be the only mechanosensitive transition, based solely on these structural models.

If mechanosensitivity were built into the pore opening, altering S6 lateral movement via mutagenesis would alter the effects of patch suction on $P_O$. However, if voltage sensor activation were additionally mechanosensitive, then voltage sensor mutagenesis would only change the response to suction but not eliminate it. We tested these ideas via site-directed mutagenesis within the S6 hinge and the voltage sensor, using NaChBac T220A as background. Most mutations we tried within the pore resulted in non-expressing or non-functional channels, but we eventually settled on I228G in the S6 hinge region (*Figure 5B*). Within the voltage sensor, we chose D93A to stabilize the sensor in the resting position (*DeCaen et al., 2009*; *Figure 5B*). We applied the same single-channel experimental paradigms to directly compare the double mutants (NaChBac T220A plus I228G or D93A) with the T220A results described above (*Figure 5C*).

The voltage sensor NaChBac T220A+D93A double mutant shifted its voltage sensitivity relative to T220A (*Figure 5D*; *Figure 5—figure supplement 1C*). However, its mechanosensitivity remained intact and followed the negative shift of voltage-dependent gating (*Figure 5D and E*). The pore NaChBac T220A+I228G double mutant channel exhibits some interesting properties. First, the channel could gate normally with voltage, like the single mutant controls (*Figure 5D*). Second, the effect of membrane tension on $P_O$ was nearly eliminated at all pressures (*Figure 5E*). Thus, at –60 mV, membrane tension increased $P_O$ by 0.096 for the NaChBac T220A mutant but only by 0.014 for NaChBac T220A+I228G, corresponding to an approximate sevenfold difference in effects between the two mutants. At –40 mV, the difference was similar (~sevenfold): 0.090 with NaChBac T220A and only 0.012 with NaChBac T220A+I228G. We could explain the small remaining effect of tension on $P_O$ in the double mutant in two ways: either there is a partial displacement of S6 during pore opening and a resulting (smaller) cross-section expansion, or there is another (weakly) mechanosensitive transition in the gating mechanism. The first possibility seems more plausible, because some degree of S6 displacement is probably necessary for channel opening, and also because NaChBac T220A+D93A maintained a tension sensitivity similar to NaChBac T220A, even though its voltage sensitivity shifted by more than –30 mV (*Figure 5D*). Overall, these mutagenesis results provide experimental evidence that strengthens our conclusion that mechanical forces interact primarily with the pore opening transition.

## Discussion

Electrically excitable cells depend on concerted efforts by VGICs to detect small changes in transmembrane voltage and amplify them to produce a wide range of action potentials (*Hodgkin and Huxley, 1952*). Some electrical organs, such as the heart, bladder, and gut, function primarily as mechanical pumps, using excitation-contraction coupling to drive muscle contractions. Cells in these pumps experience significant recurrent changes in membrane tension that can potentially affect the activity of membrane proteins, which, in turn, can affect organ function by a process called mechano-electrical feedback (*Gaub et al., 2020*; *Hao et al., 2013*; *Otway et al., 2007*; *Strege et al., 2003*). In these mechanical environments, VGICs mechanosensitivity may serve to integrate electrical (*Navarro et al., 2020*) and mechanical signals into a single control loop (*Hao et al., 2013*).

VGICs are undoubtedly mechanosensitive (*Beyder et al., 2010*; *Laitko et al., 2006*; *Morris, 2011*; *Morris and Juranka, 2007*; *Schmidt et al., 2012*; *Tabarean et al., 1999*), but the underlying mechanosensitivity mechanisms remain poorly understood, due to intrinsic structural and functional limitations. Here, we used the relatively simple bacterial voltage-gated sodium channel NaChBac as a model, because it shares crucial structural and functional elements (*Bagnéris et al., 2014*; *Ren*

*et al., 2001*) with the more complex eukaryotic voltage-gated sodium channels ($Na_V$s). We found that NaChBac (*Ren et al., 2001*) is mechanosensitive, and, impressively, the mechanosensitive responses of NaChBac closely resemble those of $Na_V$1.5 (*Figure 1*), with force increasing the peak currents and accelerating the kinetics. These effects are consistent with previous studies using macroscopic currents to examine mechanosensitivity in eukaryotic $Na_V$s (*Beyder et al., 2010*; *Morris and Juranka, 2007*) and other VGICs (*Calabrese et al., 2002*; *Gu et al., 2001*; *Schmidt et al., 2012*), which further strengthens NaChBac as a model for studying eukaryotic VGICs. In response to physiological levels of mechanical stimuli traditionally used to stimulate a mechano-gated ion channel (*Kefauver et al., 2020*), NaChBac channels substantially increased their activity in a voltage-dependent manner, in both macroscopic and single-channel preparations (*Figures 1 and 2*). Force produced a rise in the peak current evoked by depolarizing the membrane to activate the channels. However, without membrane depolarization, force alone could not open NaChBac (*Figure 1* and *Figure 2*), suggesting that mechanical force does not create new conformational states but rather impacts a single transition along the gating pathway. While whole-cell experiments proved informative, single-channel studies were required to more directly test our hypotheses.

We removed NaChBac inactivation (NaChBac T220A) (*Lee et al., 2012a*; *Lee et al., 2012b*), which allowed us to zoom in on the mechanosensitivity of voltage-dependent activation. Using the NaChBac T220A mutant, along with technical optimizations and a paired-stimulus configuration that controlled for the known resting elevated mechanical tension in patch bilayers (*Opsahl and Webb, 1994*; *Suchyna et al., 2009*), we were able to resolve sub-pA NaChBac events with mechanical stimulation (*Figures 2–5*). Patch suction modified NaChBac voltage-gating, reversibly increasing NaChBac voltage-dependent open probability ($P_O$) in a dose-dependent fashion. This effect was indeed state-dependent, suggesting that applied forces have a state-specific effect on the $Na_V$ channel, where the added mechanical energy appears to modify the energy landscape of gating but does not overcome voltage-gating (*Fowler and Sansom, 2013*; *Sigg and Bezanilla, 2003*).

To explain NaChBac mechanosensitivity, we favor a 'mechanosensitive opening' mechanism (the MSO model), rather than a 'mechanosensitive activation' (the MSA model). The MSO model features pore opening as one strongly mechanosensitive transition (*Figure 4*) and is consistent with the previous findings in $K_V$ channels, where mechanosensitivity was examined in macroscopic currents (*Schmidt et al., 2012*). Considering the simplicity of our MSO model, it is remarkable how well it could fit both whole-cell and single-channel data, under a fairly broad range of voltage and pressure values. The critical discriminator between the two competing models is the force-induced change in the macroscopic and single-channel voltage-dependent activation curves, i.e., increased maximum response and slope. The observed effects are by far better explained by the MSO model. The MSO model also accounts for the pressure-induced changes in pore opening kinetics, projecting that at maximally activating voltages, patch suction may shorten the closed state lifetimes and may destabilize the closed state. At higher pressure, patch suction may additionally lengthen the open-state lifetimes. While the structures responsible for voltage and force sensitivity may be distinct and function independently, from a kinetic mechanism standpoint, voltage and force sensitivities are state-dependent and intertwined: voltage acts on states $C_1$ through $C_5$, whereas tension acts on states $C_5$ and $O_6$. Consequently, channels must first activate by voltage before responding to tension. While simplified, this model captures the essence of the VGIC function and can apply to both prokaryotic and eukaryotic sodium channels.

By comparing the closed and open bacterial $Na_V$ crystal structures, we identified the intracellular gate as the site where the most extensive cross-section area changes occur during the transition from closed to open (*Lenaeus et al., 2017*; *McCusker et al., 2012*). The bottom halves of S6 form the intracellular gate, working like hinges on a door latched by non-covalent interactions. Functional and modeling studies support the *swinging door* model: targeting S6 residues around the pore's hinge impedes gating (*Webster et al., 2004*; *Woolfson et al., 1991*; *Zhao et al., 2004*), and pore opening leads to a physical expansion of the inner leaflet, suggesting a significant area expansion (*Beyder and Sachs, 2009*). Consistent with these studies, electrophysiology and modeling show that S6 in the pore stores the mechanical energy of gating (*Fowler and Sansom, 2013*; *Long et al., 2005*). We targeted sites separately to differentiate between the effects of force on voltage sensors from those on the pore. The S4 positively charged residues that sense voltage are stabilized in the resting state within the lipid bilayer by counterbalancing acidic (negatively charged) residues (*DeCaen et al.,*

*2009*). By mutating one of these acidic residues (D93), the half-activation and half-inactivation voltages shifted negative, but the channel maintained its responsiveness to patch pressure, confirming that voltage sensors do not significantly contribute to mechanosensitivity (*Figure 5*). Our functional data suggested that S6, forming a highly conserved component of the intracellular gate, might influence NaChBac mechanosensitivity. After many mutants turned out to be non-functional, we eventually identified and mutated a conserved hydrophobic residue, I228, located in the S6 lining the channel pore. I228G eliminated the response to pressure (*Figure 5*). The dramatic loss of I228G NaChBac mechanosensitivity suggests a loss of pressure sensitivity in the final opening step. However, it is also possible that the overall gating scheme for I228G NaChBac changed compared to its T220A background, leading to a loss of apparent dependence on the pressure-sensitive opening step. Thus, these results agree with structural and functional data showing significant in-plane area expansion during channel gating, support the *swinging door* model of VGIC pore gating, and suggest that force and voltage cooperate to gate NaChBac.

Since broad structural aspects of the intracellular gate appear conserved across VGICs, from prokaryotes to eukaryotes (*Bagnéris et al., 2014*; *Shaya et al., 2014*), we surmise that VGIC mechanosensitivity may be a generalizable, ubiquitous property, that can be observed across many families of VGICs (*Morris, 2011*; *Schmidt et al., 2012*) and across each phylum, including unicellular to complex multicellular organisms. Future studies may answer the fascinating questions of how archaic prokaryotic ion channels, including sodium channels, have developed mechanosensitivity, potentially as their earliest *sense* (*Anishkin et al., 2014*), and what role has selective pressure played in maintaining, developing, or losing this property.

How does membrane tension reach the NaChBac pore? In the *force-from-lipid* model, bilayers transduce mechanical energy directly into channel gating (*Kung, 2005*; *Martinac et al., 1990*; *Zheng et al., 2011*). For the tensed bilayer to perform work (F·d) on the channel, conformational transitions leading to the open state must associate with in-plane area expansion during the opening, and with area contraction during closing (*Sachs and Morris, 1998*). Bilayers self-assemble to minimize contact between lipid tails and water molecules. However, despite the minimization of free energy in assembled bilayers, the physical and energetic differences between phospholipid headgroups and lipid tails produce substantial intrinsic lateral forces (*Cantor, 1997*), reaching 1000 atm (*Gullingsrud and Schulten, 2004*). These lateral forces act upon the protein-lipid interface of ion channels (*Kefauver et al., 2020*; *Perozo et al., 2002b*) and have non-homogeneous effects on resident proteins through the bilayer thickness: the hydrophobic lipid core applies compression while phospholipid head groups apply tension (*Figure 6*). Specialized mechano-gated ion channels are logical candidates to take advantage of this physical arrangement, and indeed they leverage forces developed at the protein-lipid interface for their *force-from-lipid* gating (*Cox et al., 2019*; *Kefauver et al., 2020*; *Martinac et al., 1990*; *Perozo et al., 2002b*). For VGICs, both voltage sensors (*Schmidt et al., 2006*) and pore-forming structures are bathed in phospholipids (*Shaya et al., 2011*). Therefore, it is reasonable to conclude that lipids could contribute to force sensing (*Fowler and Sansom, 2013*; *Schmidt et al., 2012*), given that lipids are crucial for voltage-dependent gating (*Milescu et al., 2009*; *Schmidt*

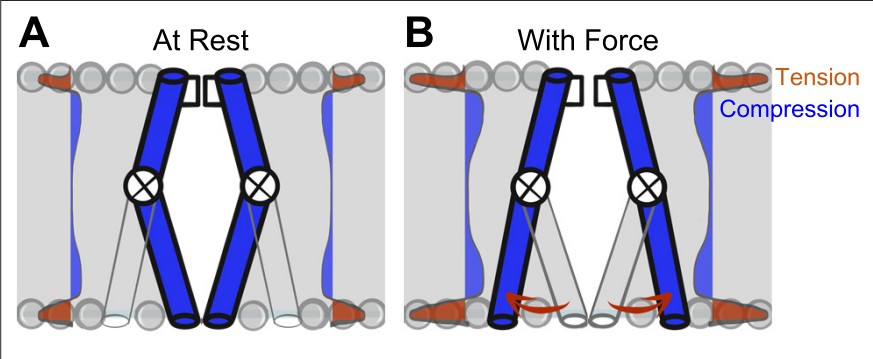

**Figure 6.** Model of voltage-gated ion channel (VGIC) mechanosensitivity. (**A**) VGIC pore is embedded in the lipid bilayer, which has an intrinsic distribution of mechanical forces even with no tension added to the system. (**B**) Mechanical stress applied to the bilayer alters the profile of bilayer forces, which destabilizes the intracellular gate and leads to intracellular pore expansion.

*et al., 2006*) and pore opening (*Fowler and Sansom, 2013*; *Morris and Juranka, 2007*; *Shaya et al., 2011*; *Zheng et al., 2011*), and lipid-permeable compounds frequently alter VGIC mechanosensitivity (*Beyder et al., 2012a*; *Cowan et al., 2022*). Further work is required to determine the energetics of intracellular pore dilation, lipid-protein interactions in VGIC mechanosensitivity, and to translate these results to eukaryotic VGICs will require technical and molecular modifications to slow down and resolve kinetics and remove inactivation.

VGIC's $P_O$-dependent mechanosensitivity has important physiologic implications, allowing $Na_V$ channels to serve as voltage-sensitive mechanosensors. Force can adjust the voltage set point for $Na_V$ channel activation and affect action potential upstroke, regulating excitability (*Conti et al., 1982*; *Conti et al., 1984*). Meanwhile, mechanosensitivity in voltage-gated potassium ($K_V$) channels (*Schmidt et al., 2012*) may serve as a mechanical brake on neuronal hyperexcitability, in a voltage-sensitive fashion (*Hao et al., 2013*). Beyond roles for VGIC mechanosensitivity in physiology, studies have uncovered patient VGIC mutations with functional disruptions in mechanosensitivity associated with diseases such as long-QT syndrome (*Banderali et al., 2010*) and irritable bowel syndrome (IBS) (*Saito et al., 2009*; *Strege et al., 2018*).

VGIC mechanosensitivity could be pharmacologically targeted in mechano-pathologies. Although specific VGIC mechanosensing inhibitors remain undeveloped, recent studies show that some amphipathic compounds that target $Na_V$ channels are effective blockers of $Na_V$ mechanosensitivity, separately from their local anesthetic mechanism (*Beyder et al., 2012a*; *Beyder et al., 2012b*; *Cowan et al., 2022*). Interestingly, the compounds' amphipathic nature is critical for function (*Beyder et al., 2012a*; *Cowan et al., 2022*), implying the channel pore's lipid-protein interface is crucial for VGIC mechanosensitivity and suggesting the intracellular gate's interaction with lipids may provide a novel pharmacologic target.

To summarize, we show here that the prokaryotic VGIC NaChBac is intrinsically mechanosensitive, and its mechanosensitivity may depend on the channel pore intracellular gate. These results offer opportunities for future studies to determine roles for $Na_V$ channel mechanosensitivity in physiology and pathophysiology and target $Na_V$ mechanosensitivity in disease.

## Materials and methods
### Cell culture

Human embryonic kidney cells (HEK293; American Type Culture Collection, Manassas, VA) were cultured in minimum essential medium (MEM, 11095–080) supplemented with 10% fetal bovine serum (FBS, 10082147) and 1% penicillin-streptomycin (15140–122, Life Technologies, Co., Grand Island, NY). Regular or Piezo1 knockout (P1KO) HEK293 cells (a kind gift from Dr. Ardem Patapoutian, Scripps Research Institute *Dubin et al., 2017*) were transfected with DNA plasmids encoding wild-type $Na_V$1.5 (variant H558/Q1077del) or wild-type or T220A NaChBac, along with GFP as a reporter, by Lipofectamine 3000 reagent (L3000-008) in OPTI-MEM medium (31985–070; Life Technologies, Co., Grand Island, NY). P1KO cells submitted to American Type Culture Collection (ATCC, Manassas, VA) for STR profiling were an exact match (eight core loci plus Amelogenin) for the Piezo1 knockout HEK293T cell line, CRL-3519. PCR testing on P1KO cells was negative for mycoplasma. Transfected cells were incubated at 37 °C for 24 hr ($Na_V$1.5) or 32 °C for 24–48 hr (WT or T220A NaChBac). Then, cells were lifted by trypsin and resuspended in NaCl Ringer's extracellular solution (composition below) before electrophysiology.

Site-directed mutagenesis was performed in the T220A NaChBac background to introduce an additional mutation, I228G or D93A, by using the QuikChange Lightning Site-Directed Mutagenesis Kit (Agilent Technologies, Santa Clara, CA). Upon verification of construct integrity and successful

**Table 3.** Primers for mutagenesis of I228G or D93A into the T220A NaChBac background.

| Mutation | Forward primer | Reverse primer |
|---|---|---|
| I228G | TCATCTTTAACTTGTTTATCGGTGTAGGCGTCAATAACGTTGAAAAAGCAGA | TCTGCTTTTTCAACGTTATTGACGCCTACACCGATAAACAAGTTAAAGATGA |
| D93A | TGGTTTGCTTTCTTAATTGTAGCCGCAGGT | ACCTGCGGCTACAATTAAGAAAGCAAACCA |

mutagenesis by DNA sequencing, either plasmid was transfected into P1KO cells for electrophysiology (*Table 3*, *Figure 5*).

## Electrophysiology

### Pipette fabrication and data acquisition

Pipettes were pulled from KG-12 or 8250 glass (King Precision Glass, Claremont, CA) for whole-cell or cell-attached patches, respectively, on a P-97 puller (Sutter Instruments, Novato, CA) and coated with HIPEC R-6101 (Dow Corning, Midland, MI). Membrane tension depends on resting pressure, applied pressure, and membrane area (*Lewis and Grandl, 2015*; *Slavchov et al., 2014*; *Suchyna et al., 2009*). The membrane area is defined by dome shape and membrane creep, factors influenced by the unique diameter and angle of each pipette tip. We kept 8250 glass pipettes within a narrow 1.2–1.5 MΩ range optimal for assessing the pressure response of channels (~4.2 µm in diameter and ~14° from wall to wall). Between pipette pairs, the heating parameter was reduced by 1–3 units in each stage of the four-stage pull to ensure that the break time fell within 2 s of the previous pair. Data were acquired with an Axopatch 200B amplifier, Digidata 1440A or 1550, and pClamp 10.6–11.2.1 software (Molecular Devices, Sunnyvale, CA).

### Recording solutions

*For whole-cell electrophysiology of WT or T220A NaChBac*, the extracellular solution was NaCl Ringer's, containing (in mM): 150 $Na^+$, 5 $K^+$, 2.5 $Ca^{2+}$, 160 $Cl^-$, 10 HEPES, 5.5 glucose, pH 7.35, 300 mmol/kg. The intracellular solution contained (in mM): 145 $Cs^+$, 5 $Na^+$, 5 $Mg^{2+}$, 125 $CH_3SO_3^-$, 35 $Cl^-$, 10 HEPES, 2 EGTA, pH 7.0, 300 mmol/kg. *For whole-cell electrophysiology of $Na_V1.5$* and *cell-attached patch-clamp of T220A NaChBac*, the bath (extracellular) solution contained (in mM): 135 $Cs^+$, 15 $Na^+$, 5 $K^+$, 2.5 $Ca^{2+}$, 160 $Cl^-$, 10 HEPES, 5.5 glucose, pH 7.35, 300 mmol/kg. The pipette solution for cell-attached patches was NaCl Ringer's, supplemented with 0.03 mM $Gd^{3+}$ to inhibit leak currents.

### Whole-cell voltage clamp

Whole-cell $Na^+$ currents from HEK293 cells heterologously expressing $Na_V1.5$ (variant H558/Q1077del) or WT or T220A NaChBac were recorded with a two-pulse protocol that tests channel activation during the first step and channel availability (steady-state inactivation) during the second step. Cells expressing $Na_V1.5$ were pulsed every 1 s from the –130 mV holding potential through –10 mV in 5 mV intervals during step 1, then immediately pulsed to –40 mV for 50 ms during step 2. $Na_V1.5$ data were sampled at 20 kHz and filtered at 5 kHz. Cells expressing NaChBac were pulsed every 4.75 s from the –120 mV holding potential through 0 mV in 10 mV intervals during step 1, then immediately pulsed to 0 mV for 50 ms (WT) or –50 mV for 400 ms (T220A) during step 2 (*Figure 1—figure supplement 1A*). NaChBac data were sampled at 2 kHz and filtered at 1 kHz.

### Cell-attached patch-clamp

P1KO cells heterologously expressing T220A NaChBac channels were held at –120 mV. To obtain single-channel events, we recorded thousands of sweeps in response to a voltage ladder protocol containing five 400 ms-long steps, from –100 mV to –20 mV in 20 mV increments, with a 3 s inter-sweep interval. Each voltage step was divided into two 200 ms-long pressure steps, from 0 mmHg to –10, –30, or –50 mmHg. Because the D93A mutant had open and closed times approximately 2–5 times longer than T220A, D93A experiments were performed with 4 s-long voltage steps and 2 s-long pressure steps. To test reversibility following pressure, the duration of each of the five voltage steps was 1 s with a 7.5 s inter-sweep interval, and pressure was applied for 500 ms (*Figure 3—figure supplement 1A*). Capacitance and passive currents were subtracted with a 1-sweep blank record, averaged from several to dozens of traces from the same or a subsequent recording in which no channel openings were observed (*Benndorf, 1994*).

### Mechanical stimulation

Mechanical stimuli were applied by shear stress to the entire cell, and by pressure clamp to membrane patches, as previously described (*Beyder et al., 2012a*; *Beyder et al., 2012b*). For whole-cell electrophysiology, shear stress was applied as the flow of extracellular solution through the 700 µL elliptical

bath chamber, for 60–90 s at 10 mL/min (**Beyder et al., 2012a**; **Strege et al., 2018**). Shear stress (1.1 dyn/cm²) was estimated by the equation $\tau = \frac{6\eta Q}{h^2 w}$, in which $\tau$ is shear stress, $\eta$ is viscosity (~1.02 cP), Q is flow rate (10 mL/min), h is solution depth (1 mm), and w is chamber width (9 mm). For cell-attached patch-clamp experiments, a negative pressure of –10 or –30 mmHg was applied by high-speed pressure clamp (HSPC-1, ALA Scientific Instruments, Farmingdale, NY) (**Besch et al., 2002**). The single-channel data were sampled at 20 kHz and low-pass filtered online at 5 kHz but for analysis were further filtered at 0.5 kHz, due to a bandwidth limitation imposed by the HSPC (**Figure 2—figure supplement 1G**). Patches are known to have non-zero resting tension (**Suchyna et al., 2009**), so we took great care to minimize the negative pressure while forming seals. The pressure clamp was set to +10 mmHg prior to the pipette entering the bath, and seals were acquired spontaneously by stepping to 0 mmHg momentarily after the pipette tip made contact with the cell membrane. Initial pipette resistance was 1–2 MΩ, and seal resistance was >10 GΩ.

## Data analysis

Data were analyzed in pClamp version 10.6 or 11.0.3 (Molecular Devices, Sunnyvale, CA), Excel 2010 (Microsoft, Redmond, WA), and SigmaPlot 12.5 (Systat Software, San Jose, CA). To estimate whole-cell conductance and the voltage of half-activation, the peak current evoked by voltage step 1 in the protocol described above was fit with a Boltzmann equation, $I_V = (V - E_{Rev}) \times G_{Max} / \left(1 + e^{\left((V - V_{1/2a})/\delta V_a\right)}\right)$, where $I_V$ is the peak current (pA/pF) at the test voltage $V$ (mV), $E_{Rev}$ is the reversal potential (mV), $G_{Max}$ is maximum conductance (nS), $V_{1/2a}$ is the half-activation voltage (mV), and $\delta V_a$ is the voltage sensitivity of activation (mV). To estimate the voltage of half-inactivation, the peak current $I_V$ evoked by voltage step 2 in the protocol was first normalized as a percentage to its maximum across all sweeps and then was fit with a Boltzmann equation, $I_V = 1 / \left(1 + e^{\left((V - V_{1/2i})/\delta V_i\right)}\right)$, where $V_{1/2i}$ is the half-inactivation voltage and $\delta V_i$ is the voltage sensitivity of inactivation. For kinetic analysis, whole-cell currents were fit to an exponential equation, $I_t = A_1 \times e^{-t/\tau_a} + A_2 \times e^{-t/\tau_i} + C$, where $\tau_a$ and $\tau_i$ are activation and inactivation time constants (ms), respectively, and $A_1$, $A_2$, and $C$ are constants.

To characterize single-channel conductance properties, all-point histograms of T220A NaChBac single-channel activity were fit with a sum of two Gaussian functions, $f(x) = A_1 \times \left(e^{-0.5 \times (x - \mu_1)^2 / \sigma_1^2}\right) / \left(\sigma_1 \times \sqrt{2\pi}\right) + A_2 \times \left(e^{-0.5 \times (x - \mu_2)^2 / \sigma_2^2}\right) / \left(\sigma_2 \times \sqrt{2\pi}\right) + C$, where $x$ is current (pA), $\mu$ and $\sigma$ represent the mean and standard deviation of the closed and open state current (pA), $A_1$ and $A_2$ are the weights of the closed and open state Gaussian components, respectively, and $C$ is baseline current. Open probability was calculated as $P_O = A_2/(A_2 + A_1)$. The response to pressure, $P_O(x) - P_O(0)$, where x stands for –10 or –30 mmHg, was obtained as the difference in $P_O$ values within the same trace. The single-channel closed and open times were calculated in QuB. Single channel time constants are expressed as means ± standard deviation (SD). Change from shear stress or pressure was considered statistically significant when $p < 0.05$ for mechano-stimulus vs. control, as determined by a two-way ANOVA with Dunnett's post-test.

## Single-channel data analysis and simulations

The analysis and simulations were done with the QuB program, the MLab edition (http://milesculabs.org/QuB.html). QuB was used to digitally low-pass filter the data at 0.5 kHz to eliminate a periodic artifact induced by the pressure clamp system (**Figure 2—figure supplement 1G**) and to extract ('idealize') the signal from the noisy data. QuB was further used to simulate the behavior of the tested NaChBac model and to calculate its properties: the voltage-activation curve at different pressures, the pressure-activation curve at different voltages, and the probability density function for closed and open dwell times, and to extract rate constants from single channel data, using the MIL algorithm that features a first-order approximation to correct for missed events (**Qin et al., 1996**).

## Na$_V$ channel model

To capture the basic properties of the NaChBac channel (homotetramer, inactivation removed), we used the simple linear kinetic scheme $C_1$-$C_2$-$C_3$-$C_4$-$C_5$-$O_6$. Each rate constant had the general expression $k = k_0 \times \exp(k_v \times V + k_p \times P)$, where $V$ is membrane potential, $P$ is patch pressure, $k_0$ is a pre-exponential factor representing the value of the rate constant at zero voltage and pressure, and $k_v$ and $k_p$ are sensitivity factors for voltage and pressure, respectively. Lack of voltage or pressure

dependence was encoded by setting $k_v$ or $k_p$ to zero. The rates along the activation pathway were in the expected 4:3:2:1 ratio (e.g. $k_{23} = 2 \times k_{45}$). The parameters of the model were tweaked by hand to match the macroscopic and single-channel data, collected within our unique experimental configuration defined above by the pipette geometry. First, we chose a set of $k_0$ preexponential parameters for the $C_5$-$O_6$ transition, to match the observed $P_O$ at saturating voltages (at –20 mV). Then, we adjusted the $k_v$ exponential parameters that describe the voltage sensitivity of the $C_1$ through $C_5$ transitions, to match the normalized macroscopic activation curve under no-shear conditions. Next, we determined the statistical distribution (average and standard deviation) of the resting potential of the single-channel patched cells—to match the voltage-dependent $P_O$ curve—which is voltage-shifted and shallower relative to the macroscopic activation curve. To generate a $P_O$ curve that takes into account the scattered and non-zero resting potential, the $P_O$ value at each voltage point was obtained by numerically integrating over the Gaussian distribution describing the resting potential. Next, we adjusted the $k_0$ preexponential parameters for the $C_1$ through $C_5$ transitions to approximately match the observed single-channel lifetimes. Finally, for the MSO model, we adjusted the $k_p$ exponential parameters describing the pressure sensitivity of the $C_5$ to $C_6$ transition, to match the $P_O$ curve under negative patch pressure. The same $k_p$ values were also used for the MSA model. The kinetic parameters used for the simulations shown in *Figure 4B–D* were the following: $k_{0,activation}$ = 800 s$^{-1}$, $k_{0,deactivation}$ = 0.1 s$^{-1}$, $k_{0,opening}$ = 70 s$^{-1}$, $k_{0,closing}$ = 55 s$^{-1}$, $k_{v,activation}$ = 0.055 V$^{-1}$, $k_{v,deactivation}$ = -0.055 V$^{-1}$, $k_{p,activation/opening}$ = -0.05 mmHg$^{-1}$, and $k_{p,deactivation/closing}$ = -0.005 mmHg$^{-1}$.

## Acknowledgements

We would like to thank Drs. Simone Mazzaferro, Steven Sine, Paul DeCaen, Fred Sachs, Mirela Milescu, Claudio Grosman, and Corrie DaCosta for their constructive suggestions, Denika Mueller for technical assistance, and Kristy Zodrow for administrative assistance. LSM acknowledges the gracious support provided by Dr. Sergei Sukharev and the University of Maryland at College Park. NIH DK052766, DK123549, AT010875.

## Additional information

### Funding

| Funder | Grant reference number | Author |
|---|---|---|
| NIDDK | DK052766 | Gianrico Farrugia Arthur Beyder |
| NIDDK | DK123549 | Arthur Beyder |
| NIH | AT010875 | Arthur Beyder |

The funders had no role in study design, data collection and interpretation, or the decision to submit the work for publication.

### Author contributions

Peter R Strege, Conceptualization, Formal analysis, Validation, Investigation, Visualization, Methodology, Writing – original draft, Writing – review and editing; Luke M Cowan, Formal analysis, Validation, Investigation, Writing – review and editing; Constanza Alcaino, Amelia Mazzone, Investigation; Christopher A Ahern, Conceptualization, Resources, Writing – review and editing; Lorin S Milescu, Conceptualization, Resources, Software, Supervision, Validation, Investigation, Methodology, Writing – review and editing; Gianrico Farrugia, Conceptualization, Funding acquisition, Validation, Project administration, Writing – review and editing; Arthur Beyder, Conceptualization, Formal analysis, Supervision, Funding acquisition, Validation, Writing – original draft, Project administration, Writing – review and editing

### Author ORCIDs

Peter R Strege [ID] http://orcid.org/0000-0003-4571-2207
Luke M Cowan [ID] http://orcid.org/0000-0002-5512-1227
Christopher A Ahern [ID] http://orcid.org/0000-0002-7975-2744

Gianrico Farrugia [ID] http://orcid.org/0000-0003-3473-5235

**Decision letter and Author response**
Decision letter https://doi.org/10.7554/eLife.79271.sa1
Author response https://doi.org/10.7554/eLife.79271.sa2

## Additional files

### Supplementary files
• MDAR checklist

### Data availability
All data generated or analysed during this study are included in the manuscript and supporting file; Source Data files have been provided for Figures 1 - 5 and Supplements to Figures 1 - 3, and 5.

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
