## [Editor Report]

This important study presents a technically impressive and carefully controlled biophysical study of the nature of the mechanosensitivity of voltage-gated sodium channels. The identification of a mechanosensitive step with little voltage sensitivity is convincing, and the proposal of a swinging door mechanism for the intracellular gate is plausible. It is expected to be of interest to scientists studying sodium channels and the physical basis of mechanosensitivity in electrophysiology.

---

## [Decision Letter]

**Decision letter after peer review:**

Thank you for submitting your article "Mechanosensitive pore opening of a prokaryotic voltage-gated sodium channel" for consideration by *eLife*. Your article has been reviewed by 3 peer reviewers, and the evaluation has been overseen by a Reviewing Editor and Kenton Swartz as the Senior Editor. The following individual involved in the review of your submission has agreed to reveal their identity: Jon Sack (Reviewer #1).

Overall, we see a conceptual advance in that Nav channels respond to pressure and shear by affecting a conformational change with little voltage dependence. As this mechanism is similar to that established for Kv channels, we suggest that the manuscript be revised to make the centerpiece core finding that this type of mechanosensitive response extends to Nav channels. We suggest that other claims and speculation including those about the structural changes, physiological relevance, and evolution be more strongly supported or substantially softened.

Essential revisions:

(1) The manuscript should make it abundantly clear that the physical mechanism proposed for the mechanosensitivity of Nav channels (a voltage-independent pore opening step) is identical to the mechanism established for a Kv channel (https://doi.org/10.1073/pnas.1204700109).

(2) Address patch tension issues as suggested by Rev #2 and #1: patch tensions may not be physiologically relevant, and the tensions produced by pressures are unknown without a calibration method.

(3) Determine whether the mechanosensitive response of Nav1.5 is reversible as suggested by Rev #3.

(4) Describe how shear force tension values were calculated.

(5) Clearly show the evidence that "… that, under tension, the closed state lifetime distribution shifts toward shorter dwell times."

(6) Resolve issues pointed out about the interpretation of I228G data by Rev #1. Ideally, present a more thorough characterization of I228G.

(7) Tone down (or remove) evolutionary speculation. See Rev #2 comments.

*Reviewer #1 (Recommendations for the authors):*

This study contains an abundance of high-quality single-channel recordings and thoughtful analyses that have convinced me that a voltage-independent step is responding to membrane tension. A weakness of the manuscript is that the story seems to go a bit beyond the cautious interpretation of results at a few points, most notably in claiming that pressure reduces closed dwell times, and an interpretation of the I228G results. Additionally, the manuscript could be improved by further discussion of the physical basis of membrane tension in a cell vs a patch, and how flow vs pressure affects tension.

Specific suggestions:

Line 43 " … mutagenesis at the hinge abolished NaChBac mechanosensitivity." As some mechanosensitivity remained, abolished seems an inappropriate word choice, and "diminished" would be more appropriate.

line 239 "with a relatively unchanged foot" State more quantitatively?

line 245 " membrane tension would not alter the voltage-dependent profile of the joint C5 and O6 occupancy"

Doesn't tension decrease occupancy of C5 thus reducing the frequency of voltage-sensitive C5->C4 transitions?

line 248 " asymmetrical shift in the activation curve at the top versus the bottom "

Explain better?

Figure 4B Why does the 0 mmHg model saturate at 1 rather than 0.6 here?

Line 287 "Although the single-channel fits are subject to inherent stochasticity (Figure 4E), they clearly show that, under tension, the closed state lifetime distribution shifts toward shorter dwell times."

I don't see the shift in the closed lifetimes in the data presented. Can the purported shift be backed up by a metric? Otherwise, the claim of a shift should be removed. Potentially the dataset could be expanded to look for stronger evidence of this shift, as it is a key prediction of the MSO model.

Figure Supp 1C I'd expect that the MSO model predicts the fold-acceleration of activation will be similar or even faster as positive voltage increases. Using a logarithmic Y-axis for time constants would help assess whether this is the case. As plotted, the time constants at -20 and -10 mV are too small to see the degree of difference.

Line 291 "If mechanosensitivity were exclusive to pore opening, preventing S6 lateral movement via mutagenesis would abolish the effects of patch suction on PO." I was unable to make sense of this. Wouldn't such a mutation also prevent the pore from gating?

I struggled to be convinced of the conclusions derived from the I228G results for 2 reasons:

A) There is only a single condition shown in Figure 5E (-40 mV) where I228G is statistically distinct from the background.

B) It is not clear what the I228G mutation does to the final opening step. I couldn't make sense of the idea that pore opening could be abolished by preventing lateral S6 movement while voltage could still gate openings of the damaged pore. However, it seems plausible that a mutation could change gating such that the mechanosensitive step becomes so biased towards allowing opening (once the voltage sensor is activated) that a change in pressure would yield little change in Po. In a simple case, one might expect such a mutation to shift the GV towards more negative voltages. Yet I228G shifts the GV towards more positive voltages. The I228G mutation does increase Po at negative voltages, but doesn't this suggest weakened voltage sensor coupling rather than a lack of lateral expansion upon pore opening?

To harden the interpretation of effects on I228G and address concerns A and B, I'd suggest a more thorough characterization of I228G (maybe similar to what was done for the T220A background) and the effects of pressure on it.

Line 391 "Physiologically relevant patch suction…" Drop the 'Physiologically relevant' qualifier? I don't see how patch suction could be physiologically relevant. A discussion of the physical coupling between pressure and patch tension seems needed in the manuscript, including a discussion of the proposal that a patched membrane is in an inherently unphysiologically-high tension environment.

Line 401: "…. increased maximum response and slope with an unchanged foot…"

Could the "foot" be described better or quantitated in some way?

Line 424: "…we left-shifted the voltage-dependence of activation but otherwise did not change mechanosensitivity, confirming that voltage sensors do not significantly contribute to mechanosensitivity…" I don't understand this logic, please clarify the physical basis of this argument.

Line 429: " While I228G did not appreciably affect voltage-gating…"

Figure 5 supplement C appears to show a change in I228G voltage gating.

Line 477: "…mechanosensitivity depends on the channel pore intracellular gate."

This remains speculation and statements could be softened to convey the speculative leap.

*Reviewer #2 (Recommendations for the authors):*

I recommended the following big picture improvements:

– A more thorough comparison to existing analogous work in other VGIC (e.g., Kv [Schmidt, …, MacKinnon PNAS 2012]) and which elements of the proposed mechano-sensing mechanism overlap, and which are novel.

– Applied pressure and shear stress are the mechanical stimuli but for pore dilation as the mechanism, lateral membrane tension is the operative quantity. Membrane tension itself is not directly measured. It is well established that patches are under non-zero tension as cited in the manuscript. It would help the reader understand the physiological significance of the findings if there was some consideration and discussion about the approximate tension regime that these experiments reflect (e.g., similar to what Piezo can sense T50 = 2.7 {plus minus} 0.1 mN/m [Lewis & Grandl *eLife* 2015] or much higher, near lytic tension)

– Related to the comment above: Line 143 states that "we could obtain and compare control and pressure data in the same cell, using test pressures relevant for mechanosensitive channel function" and then cites reference 33. The cited study shows that Piezo opens under a much smaller mechanical perturbation of 5 mmHg (poking); here we are looking at a higher pressure applied to patches that are already under tension [Opsahl & Webb Biophys J 1994].

– The evolutionary angle in the discussion should be toned down a bit. Mechanosensitivity appears to be an intrinsic feature of membrane proteins that undergo conformation changes (e.g., pore opening), which deform the lipid bilayer (thickness deformation, change in area, midplane pending [Phillips & Sens Nature 2009]). Given the existence of specialized mechano-sensor, e.g., PIEZO, that is exquisitely sensitive to lateral membrane tension [Haselwandter & Mackinnon *eLife* 2018], it could be equally well argued that adaptive changes (e.g., those that would stabilize the close state of the pore) in other ion channels may have decreased this mechanosensitivity over evolutionary timescales time to improve the fidelity for sensing voltage and decrease 'gating noise' introduced by mechanical perturbation.

– The simplicity of NaChBac is well taken, and it appears to replicate much of the mechanosensitivity observed in Nav1.5. Do mutations that affect Nav1.5 mechanosensitivity have analogous effects in NaChBac? If so, this would greatly increase the significance of this work, demonstrating that NaChBac is a disease-relevant model for Nav1.5 channelopathies.

*Reviewer #3 (Recommendations for the authors):*

(1) I suggest removing Figure 1C, and D and replacing it with Supp.1C, F. The difference current is redundant because the effect of shear stress on currents is already evident from panel B. I think that panel D would be replaced by Supplementary 1C+F, which shed more light information regarding the impact of shear stress on gating energetics.

(2) I called my attention to the lack of effect of shear stress on Nav1.5 G-V curves at the same voltage range as NaChBac channels. I believe that to make both phenomena comparable, the impact (or lack of it) on channel energetics must be discussed.

(3) I believe that to be able to extrapolate the current findings to the Nav1.5 channel and thereby highlight the physiological relevance of this work, it is necessary to evaluate the impact of pressure on Nav1.5, as shown in Figure supp 3 for NaChBac. I think that would be a good start to make crystal clear to what extent this bacterial channel is a reliable model to study mechanosensitivity in a channel that, as the authors pointed out, would otherwise remain inaccessible to these kinds of questions.

(4) It seems to me that the conclusions of the article would benefit from a discussion of the possible structural changes associated with the in-plane expansion of the pore between closed and open channel structures, and how does this correspond with the shear stress/pressure-induced ∆G you observed in your experiments.

(5) I suggest avoiding the use of "voltage dependence of activation" to identify the voltage of half-activation (V1/2) because voltage dependence is commonly associated with the apparent charge displacement (zd).

[Editors' note: further revisions were suggested prior to acceptance, as described below.]

Thank you for resubmitting your work entitled "Mechanosensitive pore opening of a prokaryotic voltage-gated sodium channel" for further consideration by *eLife*. Your revised article has been evaluated by Kenton Swartz (Senior Editor) and a Reviewing Editor.

The thoughtful revisions and additional experiments have improved the manuscript. However, several of the requested essential revisions require further attention:

"(2) Address patch tension issues as suggested by Rev #2 and #1: patch tensions may not be physiologically relevant, and the tensions produced by pressures are unknown without a calibration method. "

The details of pressures applied during the patch-clamp seal are an improvement to the manuscript, as are the edits to distinguish further between pressure and tension. However, the manuscript does not address the issue of a calibration method: ∆Tension/∆Pressure depends on the membrane area, and is thus expected to vary from patch to patch. Due to this issue, the pressure sensitivity in the Markov chain modules applies only to this specific experimental configuration.

Suggestions: Clearly acknowledge these issues in the manuscript. Describe the expected relationship between pressure and tension with membrane varying from patch to patch and cite literature that addresses this issue. Further details about pipettes would helpful as well. Initial pipette resistance was 1-2 MΩ: was variance in pipette geometry documented? More detail on the pipette resistances, and ideally tip diameter and bevel, could be given, and how consistent the tip structure was for each mutant, to address concerns related to variable scaling between pressure and tension. Was there a correlation between initial pipette resistance, the timing of the seal, and the pressure response of channels?

"(5) Clearly show the evidence that "… that, under tension, the closed state lifetime distribution shifts toward shorter dwell times."

The closed dwell distribution in Figure 4E still does not seem to provide compelling support for the conclusion that pressure alters microscopic opening rates. Although the manuscript now reports that fits with and without pressure show ~12% different opening rates, the distributions themselves do not show an obvious divergence. SEMs are given for the different opening rates but it is unclear what the SEMs refer to. There are no apparent replicates.

Suggestion: Eliminate the claim of experimental evidence for channel opening accelerated by pressure, or provide stronger evidence for the claim by providing sufficient replicates and a detailed description of the analysis methods.

"(6) Resolve issues pointed out about the interpretation of I228G data by Rev #1. Ideally, present a more thorough characterization of I228G. "

I228G results now show nicely that the gating of the mutant is less pressure-sensitive. This is a very interesting, well-substantiated result! However, the logic remains unclear concerning conclusions derived from the S6 mutant which lessens mechanosensitivity. It seems that multiple explanations could account for the I228G results, even if a pressure-sensitive gating change was no longer detected. It remains unclear how the I228G could gate (relatively normally) if the opening conformational change (S6 displacement) no longer displaces.

Suggestion: Mention that multiple interpretations of the diminished pressure sensitivity of I228G are plausible: (A) The gating scheme has changed such that the pressure-sensitive step no longer measurably alters the gating process. (B) The pressure-sensitive step no longer occurs. (C) The pressure-sensitive step is longer pressure sensitive (the current interpretation).

---

## [Author Response]

Essential revisions:1) The manuscript should make it abundantly clear that the physical mechanism proposed for the mechanosensitivity of Nav channels (a voltage-independent pore opening step) is identical to the mechanism established for a Kv channel (https://doi.org/10.1073/pnas.1204700109).

We appreciate this point and thus have cited this landmark study extensively throughout our manuscript. This Kv channel paper used macroscopic currents with pressure stimuli to propose a voltage-gated channel mechanosensitivity mechanism that centered around a mechanosensitive pore. As the reviewers and editors pointed out, our study matches these findings, but in Nav channels and, very importantly, using not only macroscopic but also single channel analysis. We also made other discoveries and provide new tools in this study. Nevertheless, we have made every effort to appropriately acknowledge the important contributions of the MacKinnon study.

(2) Address patch tension issues as suggested by Rev #2 and #1: patch tensions may not be physiologically relevant, and the tensions produced by pressures are unknown without a calibration method.

We acknowledge the existence of a non-zero resting pressure in patches, and we mention this fact in several places in the manuscript. We cannot measure the resting pressure, but we took great care to (1) minimize the pressures required to form patches (all obtained at 0 mmHg) and (2) always use paired comparisons, with control (no added pressure) vs. added pressure, allowing us to make direct comparisons. The pressures we used in the single-channel experiments align with the values used in the literature, as cited in the manuscript. We added further clarification to the manuscript.

3) Determine whether the mechanosensitive response of Nav1.5 is reversible as suggested by Rev #3.

The experiments in this study build on knowledge gained from our previous work, where we demonstrated the reversibility of Na_V_1.5 mechanosensitivity in both whole-cell preps exposed to shear stress and single-channel patches subjected to pressure pulses (Saito et al. *Am J Physiol Gastrointest Liver Physiol* 2009, Beyder et al. *Circulation* 2012, Strege et al. *Channels* 2019). One caveat we discovered is that pressure pulses must be relatively short to prevent patch creep (Beyder et al. *J Physiol* 2010). We have all the supporting citations in the manuscript.

We also added Figure 3 Supplement 2, which demonstrates the reversible response of Nav1.5 currents to shear stress in n = 24 cells.

4) Describe how shear force tension values were calculated.

We added the following sentence to the Methods:

“Shear stress (1.1 dyn/cm^2^) was estimated by the equation τ=6ηQh2w , in which τ is shear stress, η is viscosity (~1.02 cP), Q is the flow rate (10 mL/min), h is solution depth (1 mm), and w is chamber width (9 mm).”

5) Clearly show the evidence that "… that, under tension, the closed state lifetime distribution shifts toward shorter dwell times."

We address this issue in depth below in response to Reviewer #1. Briefly, as suggested by the reviewer, we quantified the changes in closed state lifetimes to be ~12% shorter with -10 mmHg pressure, while open state lifetimes were unchanged. We have added this analysis to the manuscript.

(6) Resolve issues pointed out about the interpretation of I228G data by Rev #1. Ideally, present a more thorough characterization of I228G.

We appreciate and thank the reviewers and editors for raising this important point. In response, we substantially expanded the I228G NaChBac dataset (Figure 5 D-E, Supplement 2) by increasing the number of patches at -10 mmHg and adding single-channel data at -30 and -50 mmHg, as well as macroscopic data at -30 mmHg, all across the relevant voltage range. These new data are consistent with our initial interpretation, with the new data clearly showing that I228G NaChBac loses mechanosensitivity at all pressures and across all tested voltages. We modified the Discussion section accordingly.

7) Tone down (or remove) evolutionary speculation. See Rev #2 comments.

As requested, we have significantly toned down speculations about evolution.

Reviewer #1 (Recommendations for the authors):This study contains an abundance of high-quality single-channel recordings and thoughtful analyses that have convinced me that a voltage-independent step is responding to membrane tension. A weakness of the manuscript is that the story seems to go a bit beyond the cautious interpretation of results at a few points, most notably in claiming that pressure reduces closed dwell times, and an interpretation of the I228G results. Additionally, the manuscript could be improved by further discussion of the physical basis of membrane tension in a cell vs a patch, and how flow vs pressure affects tension.

We appreciate the constructive feedback and address each point below.

Specific suggestions:Line 43 " … mutagenesis at the hinge abolished NaChBac mechanosensitivity." As some mechanosensitivity remained, abolished seems an inappropriate word choice, and "diminished" would be more appropriate.

We agree and have made this change.

line 239 "with a relatively unchanged foot" State more quantitatively?Doesn't tension decrease occupancy of C5 thus reducing the frequency of voltage-sensitive C5->C4 transitions?line 248 " asymmetrical shift in the activation curve at the top versus the bottom "Explain better?

The "unchanged foot" supports the idea of mechanosensitivity associated with a voltageinsensitive transition (the pore opening). Unfortunately, the "foot" of the curve does not have a direct relationship with any specific parameter in the Boltzmann equation, although it could be empirically characterized as the voltage where Po reaches a minimal threshold.

Considering the comment raised by the reviewer, we decided it is best to eliminate references to the notions of "foot" and "asymmetrical shift," as a simple visual inspection of the data should be enough to understand what happens to the activation curves when we add pressure.

line 245 " membrane tension would not alter the voltage-dependent profile of the joint C5 and O6 occupancy"

The reviewer is correct. We revised the explanation given in the text.

Figure 4B Why does the 0 mmHg model saturate at 1 rather than 0.6 here?

This figure compares the whole-cell Na^+^ conductance, increased by shear stress, and the model prediction. Because we do not know what the actual Po is for these macroscopic data, we normalized the model prediction to 1.

Line 287 "Although the single-channel fits are subject to inherent stochasticity (Figure 4E), they clearly show that, under tension, the closed state lifetime distribution shifts toward shorter dwell times."I don't see the shift in the closed lifetimes in the data presented. Can the purported shift be backed up by a metric? Otherwise, the claim of a shift should be removed. Potentially the dataset could be expanded to look for stronger evidence of this shift, as it is a key prediction of the MSO model.

We appreciate an opportunity to clarify. The observed pressure-induced increase in Po must be explainable by a change in some data parameter. Thus, we analyzed the single-channel data obtained from the T220A mutant at -20 mV using the model-based, maximum likelihood method implemented in the QuB software (the MIL algorithm described in Qin et al. *Biophys J.* 1996). This method is better than fitting exponentials because it accounts for missed events. We used a CO model, which is justified by the saturation of the G-V curve at -20 mV, where the channel is pushed into the last two kinetic states of the linear model. Thus, when the patch pressure is changed from 0 to -10 mmHg, the CàO rate constant changes from 127.7 ±5.6 s^-1^ to 142.4 ±6.1 s^-1^ (12%). This increase in the opening rate constant reduces the average duration of closed intervals, shifting the closed dwell time distribution toward shorter times. In contrast, the O→C rate constant remains virtually unchanged (48.6 ±2.3 s^-1^ vs. 48.4 ±2.3 s^-1^ [-0.5%]). We added this quantification to the revised manuscript.

Figure Supp 1C I'd expect that the MSO model predicts the fold-acceleration of activation will be similar or even faster as positive voltage increases. Using a logarithmic Y-axis for time constants would help assess whether this is the case. As plotted, the time constants at -20 and -10 mV are too small to see the degree of difference.

We appreciate the reviewer's suggestion and have rescaled the time constants in Figure 1 Supp 1C on a logarithmic scale. The time constant of activation for WT and T220A NaChBac does appear to accelerate with increasing voltages, supporting the MSO model (Author response image 1). However, we remain cautious in comparing the fold change in time constants across the voltage range because the primary source of error in whole-cell

recordings becomes distinctly different toward each extreme (signal-to-noise ratio toward negative voltages *vs.* preset sampling rate toward positive voltages). Therefore, we elected not to include this analysis in the manuscript.

**Author response image 1. sa2fig1:** 

Line 291 "If mechanosensitivity were exclusive to pore opening, preventing S6 lateral movement via mutagenesis would abolish the effects of patch suction on PO." I was unable to make sense of this. Wouldn't such a mutation also prevent the pore from gating?

Although we identify this S6 lateral displacement as likely happing during the opening, we cannot be sure that this displacement is an absolute requirement for the pore to open. As it happens, the channel still opens, even though the displacement is reduced in the mutant. We nuanced the statement to "if mechanosensitivity were built into pore opening, altering S6 lateral movement via mutagenesis would alter the effects of patch suction on P_O_."

I struggled to be convinced of the conclusions derived from the I228G results for 2 reasons:A) There is only a single condition shown in Figure 5E (-40 mV) where I228G is statistically distinct from the background.

We agree with the reviewer. Therefore, we significantly expanded the I228G dataset by increasing *n* at -10 mmHg and adding new data at -30 and -50 mmHg, as we did for the T220A construct. We present the new data in Figures 5D and E, and clearly show that I228G lacks mechanosensitivity across the tested voltage range.

B) It is not clear what the I228G mutation does to the final opening step. I couldn't make sense of the idea that pore opening could be abolished by preventing lateral S6 movement while voltage could still gate openings of the damaged pore. However, it seems plausible that a mutation could change gating such that the mechanosensitive step becomes so biased towards allowing opening (once the voltage sensor is activated) that a change in pressure would yield little change in Po. In a simple case, one might expect such a mutation to shift the GV towards more negative voltages. Yet I228G shifts the GV towards more positive voltages. The I228G mutation does increase Po at negative voltages, but doesn't this suggest weakened voltage sensor coupling rather than a lack of lateral expansion upon pore opening?

The reviewer raises good points. However, although the I228G mutation is supposed to reduce the S6 lateral displacement, it could also alter some different steps in the gating mechanism, with somewhat unpredictable effects. Losing mechanosensitivity does not necessarily mean easier opening, but the opposite may be true, depending on how and which rates change upon mutation, which could explain the rightward shift of the I228G G-V curve. We could only speculate on these points, but the one fact that we can be sure of is that mechanosensitivity is much reduced in I228G, in marked contrast with the D93A mutant, which maintains mechanosensitivity but exhibits drastically altered voltage sensitivity.

To harden the interpretation of effects on I228G and address concerns A and B, I'd suggest a more thorough characterization of I228G (maybe similar to what was done for the T220A background) and the effects of pressure on it.

As suggested, we have expanded our data to include the effects of pressure on I228G at multiple pressures and voltages and found that the mean change in Po (∆Po) for T220A Po was pressure dependent (>10%) at -60 and -40 mV for all pressures, while the ∆Po for I228G was not above 5% at any voltage or pressure (Figure 5, Supplement 2A). Furthermore, we also obtained macroscopic patch data showing that in patches with ≥3 channels, -30 mmHg pressure increased T220A currents by 14.9±5.3% (n=10), while I228G currents by just 2.0±2.0% (n=7, *P*<0.05 to T220A by a 2-tailed nonparametric t-test). Thus, these data are consistent with our initial conclusions, and we have them added to the manuscript.

Line 391 "Physiologically relevant patch suction…" Drop the 'Physiologically relevant' qualifier? I don't see how patch suction could be physiologically relevant. A discussion of the physical coupling between pressure and patch tension seems needed in the manuscript, including a discussion of the proposal that a patched membrane is in an inherently unphysiologically-high tension environment.

We used patch pressures typical for gating mechanosensitive ion channels, and thus we considered them physiologically relevant. Nevertheless, the reviewer's point is fair. We dropped the qualifier, as suggested, and added a comparison between membrane tensions in the patch versus the whole cell.

Line 401: "…. increased maximum response and slope with an unchanged foot…"Could the "foot" be described better or quantitated in some way?

We agree with this comment, and as discussed above, we decided to eliminate all references to the "foot."

Line 424: "…we left-shifted the voltage-dependence of activation but otherwise did not change mechanosensitivity, confirming that voltage sensors do not significantly contribute to mechanosensitivity…" I don't understand this logic, please clarify the physical basis of this argument.

We modified this sentence to clarify that even though the D93A NaChBac V_1/2_ shifted negative, it was still responsive to pressure in the patch.

Line 429: " While I228G did not appreciably affect voltage-gating…"Figure 5 supplement C appears to show a change in I228G voltage gating.

We have removed this interpretation to reflect the data in the supplement better.

Line 477: "…mechanosensitivity depends on the channel pore intracellular gate."This remains speculation and statements could be softened to convey the speculative leap.

We agree with the reviewer. We have softened this statement to say that "mechanosensitivity may depend on the channel pore intracellular gate."

Reviewer #2 (Recommendations for the authors):I recommended the following big picture improvements:– A more thorough comparison to existing analogous work in other VGIC (e.g., Kv [Schmidt, …, MacKinnon PNAS 2012]) and which elements of the proposed mechano-sensing mechanism overlap, and which are novel.

We cited MacKinnon's PNAS paper as an example of Kv channel mechanosensitivity, and we expanded the paragraph in the introduction to mention the study's findings specifically. In the Discussion, we compare Na_V_s and other VGICs, and state that our model is consistent with the MacKinnon study.

– Applied pressure and shear stress are the mechanical stimuli but for pore dilation as the mechanism, lateral membrane tension is the operative quantity. Membrane tension itself is not directly measured. It is well established that patches are under non-zero tension as cited in the manuscript. It would help the reader understand the physiological significance of the findings if there was some consideration and discussion about the approximate tension regime that these experiments reflect (e.g., similar to what Piezo can sense T50 = 2.7 {plus minus} 0.1 mN/m [Lewis & Grandl eLife 2015] or much higher, near lytic tension)– Related to the comment above: Line 143 states that "we could obtain and compare control and pressure data in the same cell, using test pressures relevant for mechanosensitive channel function" and then cites reference 33. The cited study shows that Piezo opens under a much smaller mechanical perturbation of 5 mmHg (poking); here we are looking at a higher pressure applied to patches that are already under tension [Opsahl & Webb Biophys J 1994].

The reviewer is correct in pointing to the established knowledge that patches, even at rest, are at some underlying non-zero tension. Therefore, we have carefully designed our experiments to (1) form seals at minimal pressures (0 mmHg) and (2) have in-patch controls to compare the effects of applied pressure directly. Unfortunately, the need for very low noise recordings prevented us from performing video recordings of patches during the application of force. As the reviewer requested, we modified the citations, removing the Coste reference and replacing it with the Opsahl reference.

– The evolutionary angle in the discussion should be toned down a bit. Mechanosensitivity appears to be an intrinsic feature of membrane proteins that undergo conformation changes (e.g., pore opening), which deform the lipid bilayer (thickness deformation, change in area, midplane pending [Phillips & Sens Nature 2009]). Given the existence of specialized mechano-sensor, e.g., PIEZO, that is exquisitely sensitive to lateral membrane tension [Haselwandter & Mackinnon eLife 2018], it could be equally well argued that adaptive changes (e.g., those that would stabilize the close state of the pore) in other ion channels may have decreased this mechanosensitivity over evolutionary timescales time to improve the fidelity for sensing voltage and decrease 'gating noise' introduced by mechanical perturbation.

We agree with the reviewer, and we toned down the evolutionary speculations.

– The simplicity of NaChBac is well taken, and it appears to replicate much of the mechanosensitivity observed in Nav1.5. Do mutations that affect Nav1.5 mechanosensitivity have analogous effects in NaChBac? If so, this would greatly increase the significance of this work, demonstrating that NaChBac is a disease-relevant model for Nav1.5 channelopathies.

This is a great suggestion but, unfortunately, outside the scope of the current study, which has taken us over five years to complete.

Reviewer #3 (Recommendations for the authors):1) I suggest removing Figure 1C, and D and replacing it with Supp.1C, F. The difference current is redundant because the effect of shear stress on currents is already evident from panel B. I think that panel D would be replaced by Supplementary 1C+F, which shed more light information regarding the impact of shear stress on gating energetics.

We carefully considered the reviewer's suggestion but ultimately decided to keep the current layout because we would like to emphasize the increase in G/Gmax, which is the signature parameter that changes with shear stress and becomes the critical difference between the MSO and MSA models.

2) I called my attention to the lack of effect of shear stress on Nav1.5 G-V curves at the same voltage range as NaChBac channels. I believe that to make both phenomena comparable, the impact (or lack of it) on channel energetics must be discussed.

The fast activation and inactivation kinetics make direct comparisons between NaV1.5 and NaChBac very difficult. We lean on the impressive similarities between the mechanosensitive responses by both channels. We have adjusted the Discussion section to note that close comparison between NaV1.5 and NaChBac will require further simplification of NaV1.5 function and dramatic technical improvements.

3) I believe that to be able to extrapolate the current findings to the Nav1.5 channel and thereby highlight the physiological relevance of this work, it is necessary to evaluate the impact of pressure on Nav1.5, as shown in Figure supp 3 for NaChBac. I think that would be a good start to make crystal clear to what extent this bacterial channel is a reliable model to study mechanosensitivity in a channel that, as the authors pointed out, would otherwise remain inaccessible to these kinds of questions.

The reviewer brings up a fair point. Removing fast inactivation from Na_V_1.5 would allow us to evaluate NaV1.5 mechanosensitivity appropriately. We previously tried inactivation-removed Na_V_1.5 channels (e.g., WCW) but have encountered technical problems. We hope to solve these issues and be able to present the findings in future work. We have softened the generalizations in the Discussion.

4) It seems to me that the conclusions of the article would benefit from a discussion of the possible structural changes associated with the in-plane expansion of the pore between closed and open channel structures, and how does this correspond with the shear stress/pressure-induced ∆G you observed in your experiments.

The reviewer is asking an intriguing question. Unfortunately, this type of analysis is beyond our capabilities. Still, as suggested, we have alluded to such an evaluation in future studies in the Discussion.

5) I suggest avoiding the use of "voltage dependence of activation" to identify the voltage of half-activation (V1/2) because voltage dependence is commonly associated with the apparent charge displacement (zd).

We replaced "voltage dependence of activation" with "half-point of steady-state activation."

[Editors' note: further revisions were suggested prior to acceptance, as described below.]

The thoughtful revisions and additional experiments have improved the manuscript. However, several of the requested essential revisions require further attention:"(2) Address patch tension issues as suggested by Rev #2 and #1: patch tensions may not be physiologically relevant, and the tensions produced by pressures are unknown without a calibration method. "The details of pressures applied during the patch-clamp seal are an improvement to the manuscript, as are the edits to distinguish further between pressure and tension. However, the manuscript does not address the issue of a calibration method: ∆Tension/∆Pressure depends on the membrane area, and is thus expected to vary from patch to patch. Due to this issue, the pressure sensitivity in the Markov chain modules applies only to this specific experimental configuration.Suggestions: Clearly acknowledge these issues in the manuscript. Describe the expected relationship between pressure and tension with membrane varying from patch to patch and cite literature that addresses this issue. Further details about pipettes would helpful as well. Initial pipette resistance was 1-2 MΩ: was variance in pipette geometry documented? More detail on the pipette resistances, and ideally tip diameter and bevel, could be given, and how consistent the tip structure was for each mutant, to address concerns related to variable scaling between pressure and tension. Was there a correlation between initial pipette resistance, the timing of the seal, and the pressure response of channels?

We appreciate this point. We have spent much time optimizing pipette geometry. Single-channel data for this paper were collected across 5 years. Several factors can contribute to variability, including heating filament and fire-polishing. We only used 1-2 MΩ pipettes, but we did not document the initial pipette resistance, pipette geometry, or timing of the seal to be able to reconstruct a test looking for a correlation with pressure response in each experiment. In a present-day check of this method, we found that pipettes that we would use for successful patch experiments were within a narrow 1.2-1.5 MΩ range (1.31±0.02 MΩ), measuring 4.23±0.06 µm in diameter and 14.03±0.30° from wall-to-wall (n = 16 pipette tips). We have added these methodological details to the manuscript.

As requested, we also acknowledge in Methods that (1) the change in tension resulting from applied pressure depends on the membrane area (Suchyna et al., *Biophys J* 2009; Slachov et al., *J Phys Chem B* 2014; Lewis and Grandl *ELife* 2015) and (2) the pressure sensitivity in the Markov model applies only to our experimental configuration, in terms of actual numbers, while model discrimination is general.

"(5) Clearly show the evidence that "… that, under tension, the closed state lifetime distribution shifts toward shorter dwell times."The closed dwell distribution in Figure 4E still does not seem to provide compelling support for the conclusion that pressure alters microscopic opening rates. Although the manuscript now reports that fits with and without pressure show ~12% different opening rates, the distributions themselves do not show an obvious divergence. SEMs are given for the different opening rates but it is unclear what the SEMs refer to. There are no apparent replicates.Suggestion: Eliminate the claim of experimental evidence for channel opening accelerated by pressure, or provide stronger evidence for the claim by providing sufficient replicates and a detailed description of the analysis methods.

We have taken the second suggested approach – “to strengthen the claim by providing sufficient replicates and a detailed description of the analyses methods.” We have added new data and have expanded the description of the methods, as we detail below.

Methods. We followed a standard procedure in the field for the single-channel kinetic analysis, consisting of two steps: (1) data “idealization,” where a dwell time sequence is generated by classifying each data point as coming from a closed or open channel, and (2) rate constant estimation, where a set of rate constants is obtained that maximizes the likelihood of the dwell time sequence. To “idealize” the data, we used the “Segmental K-Means” algorithm or the half-amplitude threshold method, as implemented in QuB software and pClamp. To estimate rate constants, we used QuB’s “Maximum Interval Likelihood” method (MIL), which features a correction for missed events due to finite bandwidth. The error should indicate standard deviation, not SEMs, calculated in QuB from n = 124 traces and 10 patches at -10 mmHg and n = 23 traces and 3 patches at -50 mmHg (see below). We apologize for the oversight and have corrected the methods.

The maximum likelihood approach is superior to fitting dwell time histograms, which do not account for missed events and cannot distinguish well between kinetic mechanisms. Nevertheless, we showed the dwell time histograms, together with the pdf curves calculated by the maximum likelihood algorithm (accounting for missed events), to give the reader a visual sense of data statistics and model fitness.

Data. The increase in p_Open_ with pressure is very robust and can be observed in both single channel (Figure 2, 3, 3 Suppl 1) and whole cell (Figure 4B, 5 Suppl 2) data under a broad range of voltages. At extreme depolarizing voltages (i.e., –20 mV), the channel can be assumed to flicker between the last two states in the kinetic scheme, closed and open. Under these conditions, p_Open_ can be calculated with the equation k_Open_/(k_Open_ + k_Close_). Thus, a change in p_Open_ with pressure can only be explained by changes in k_Open_, k_Close_, or both. However, a change in k_Open_ or k_Close_ by a factor as small as 1.5 may be sufficient to explain the observed change in p_Open_ of ~0.1, upon application of -10 mmHg patch pressure. Such a small change in rate constants would entail a correspondingly small shift in the dwell time histograms, which would be difficult to detect in the presence of stochastic fluctuations, and especially when the histogram must be plotted on a logarithmic time scale that spans several decades.

To make it more obvious to the readers that patch pressure alters the rate constants of the pore opening transition, we took your advice and added new data collected at higher pressures (-50 mmHg), where the change in rate constants was more apparent. To eliminate the potential confusion raised by dwell time histograms, we have now graphed the estimated k_Open_ and k_Close_ rate constants (Figure 4E). The change in k_Open_ is substantial and significant. Interestingly, at high pressures, we also saw small changes in k_Close_. Overall, these changes in the pore opening transition quantitatively explain the increase in p_Open_ with pressure.

"(6) Resolve issues pointed out about the interpretation of I228G data by Rev #1. Ideally, present a more thorough characterization of I228G. "I228G results now show nicely that the gating of the mutant is less pressure-sensitive. This is a very interesting, well-substantiated result! However, the logic remains unclear concerning conclusions derived from the S6 mutant which lessens mechanosensitivity. It seems that multiple explanations could account for the I228G results, even if a pressure-sensitive gating change was no longer detected. It remains unclear how the I228G could gate (relatively normally) if the opening conformational change (S6 displacement) no longer displaces.Suggestion: Mention that multiple interpretations of the diminished pressure sensitivity of I228G are plausible: (A) The gating scheme has changed such that the pressure-sensitive step no longer measurably alters the gating process. (B) The pressure-sensitive step no longer occurs. (C) The pressure-sensitive step is longer pressure sensitive (the current interpretation).

We intended to phrase the I228G discussion to acknowledge that (as pointed out) the gating mechanism of I228G gating is unknown and that our interpretations are not the only plausible explanations. At your suggestion, we have added, “The dramatic loss of I228G NaChBac mechanosensitivity suggests a loss of pressure sensitivity in the final opening step. However, it is also possible that the overall gating scheme for I228G NaChBac changed compared, leading to a loss of the pressure-sensitive opening step.”